# Control of 3′ splice site selection by the yeast splicing factor Fyv6

Katherine A Senn[1†], Karli A Lipinski[2†], Natalie J Zeps[1], Amory F Griffin[1], Max E Wilkinson[3*‡, §], Aaron A Hoskins[1,2*]

[1]Department of Biochemistry, University of Wisconsin-Madison, Madison, United States; [2]Department of Chemistry, University of Wisconsin-Madison, Madison, United States; [3]MRC Laboratory of Molecular Biology, Cambridge, United Kingdom

*For correspondence:
mwilkin@mit.edu (MEW);
ahoskins@wisc.edu (AAH)

[†]These authors contributed equally to this work

Present address: [‡]Broad Institute of MIT and Harvard, Cambridge, United States; [§]McGovern Institute for Brain Research, Massachusetts Institute of Technology, Cambridge, United States

## eLife Assessment

This **important** study addresses how 3' splice site choice is modulated by the conserved spliceosome-associated protein Fyv6. The authors provide **compelling** evidence that Fyv6 functions to enable selection of 3' splice sites distal to a branch point and in doing so antagonizes more proximal, suboptimal 3' splice sites.

**Abstract** Pre-mRNA splicing is catalyzed in two steps: 5′ splice site (SS) cleavage and exon ligation. A number of proteins transiently associate with spliceosomes to specifically impact these steps (first and second step factors). We recently identified Fyv6 (FAM192A in humans) as a second step factor in *Saccharomyces cerevisiae*; however, we did not determine how widespread Fyv6's impact is on the transcriptome. To answer this question, we have used RNA sequencing (RNA-seq) to analyze changes in splicing. These results show that loss of Fyv6 results in activation of non-consensus, branch point (BP) proximal 3′ SS transcriptome-wide. To identify the molecular basis of these observations, we determined a high-resolution cryo-electron microscopy (cryo-EM) structure of a yeast product complex spliceosome containing Fyv6 at 2.3 Å. The structure reveals that Fyv6 is the only second step factor that contacts the Prp22 ATPase and that Fyv6 binding is mutually exclusive with that of the first step factor Yju2. We then use this structure to dissect Fyv6 functional domains and interpret results of a genetic screen for *fyv6Δ* suppressor mutations. The combined transcriptomic, structural, and genetic studies allow us to propose a model in which Yju2/Fyv6 exchange facilitates exon ligation and Fyv6 promotes usage of consensus, BP distal 3′ SS.

## Introduction

Precursor messenger RNA (pre-mRNA) splicing is catalyzed by a large macromolecular complex called the spliceosome. Spliceosomes are composed of five small nuclear ribonucleoproteins (snRNPs), each composed of a small nuclear RNA (snRNA; U1, U2, U4, U5, and U6) and several different protein splicing factors. The snRNPs and dozens of other proteins assemble de novo at each intron to form spliceosomes. Spliceosomes are highly dynamic and form different complexes as proteins and snRNAs join and leave. Many of these complexes have been characterized biochemically, genetically, and by cryo-electron microscopy (cryo-EM) (*Plaschka et al., 2019*). The splicing reaction itself is carried out in two sequential transesterification reactions (*Figure 1A*). First, the 5′ splice site (5′ SS) is cleaved during formation of an intron lariat (first step). Second, the intron lariat is removed simultaneously with exon ligation at the 3′ SS (second step).

The integrity of the genetic information contained within a pre-mRNA depends on correct identification of the 5′ and 3′ SS by the splicing machinery since a single nucleotide shift in either site could

destroy a protein reading frame. In yeast, several DExD/H-box ATPases function to limit usage of suboptimal SS by enhancing splicing fidelity (*Chung et al., 2023*; *Semlow and Staley, 2012*). Two of these ATPases, Prp16 and Prp22, impact the fidelity of the first and second catalytic steps, respectively. In addition to these roles, Prp16 is also required for remodeling of the spliceosome to permit the first to second step transition (*Schwer and Guthrie, 1992*), and Prp22 is essential for releasing the mRNA product (*Company et al., 1991*). While Prp16 does not need to be present during the first step (*Chung et al., 2023*), Prp22 is required to be present for exon ligation if the branch point (BP) to 3′ SS distance is ≥21 nucleotides (nt) (*Schwer and Gross, 1998*). Despite many biochemical, single molecule, and structural studies, how spliceosomes promote usage of BP distal, Prp22-dependent 3′ SS has remained elusive.

Identification of the correct 3′ SS is an especially challenging problem given that the consensus sequence in *Saccharomyces cerevisiae* (yeast) and humans is just three nucleotides (YAG, Y=U or C). Cryo-EM structures of spliceosome product (P) complexes have revealed how these nucleotides can be recognized within the spliceosome active site (*Bai et al., 2017*; *Fica et al., 2019*; *Liu et al., 2017*; *Wilkinson et al., 2017*; *Zhang et al., 2019*). The AG 3′ SS dinucleotide is recognized by the formation of non-Watson-Crick base-pairing interactions with the 5′ SS +1G and BP adenosine, while the pyrimidine at the –3 position of the 3′ SS is recognized by the Prp8 protein. Given this short consensus sequence, multiple different splicing factors also help to ensure that the correct 3′ SS is utilized, including Cwc21 and the second step factors Slu7, Prp18, and Prp22 (*Crotti et al., 2007*; *Frank and Guthrie, 1992*; *Gautam et al., 2015*; *Kawashima et al., 2014*; *Roy et al., 2023*; *Semlow et al., 2016*). It is believed that Slu7 and Prp18 help to stabilize 3′ SS docking to the active site and that this contributes to second step efficiency. It has been proposed that Prp22 antagonizes 3′ SS docking to permit sampling of different 3′ SS to ensure that optimal sequences are used as a proofreading mechanism (*Mayas et al., 2006*; *Semlow et al., 2016*). Importantly, identification of the correct 3′ SS depends not just on identification of the YAG sequence but also involves choosing which YAG sequence to use. Prp18 appears to aid in selection of BP distal 3′ SS and avoidance of BP proximal (and often nonconsensus) 3′ SS. This observation could be due to Prp18 imposing a BP to 3′ SS distance constraint, enforcing use of canonical YAG 3′ SS sequences, or a combination of the two activities.

Recently, Fyv6 was identified as a novel second step factor in yeast (*Lipinski et al., 2023*). Fyv6 is a homolog of the human protein FAM192A, which was identified in pre-C* structures of the human spliceosome (*Zhan et al., 2022*). While the impact of FAM192A on human spliceosome activity has not yet been well characterized, loss of Fyv6 in yeast decreases second step splicing efficiency in vitro and results in use of an alternative 3′ SS in the *SUS1* pre-mRNA in vivo. It is likely that unassigned cryo-EM densities in yeast C* (just prior to exon ligation) and P complexes correspond to Fyv6 since these densities are in analogous locations to that for FAM192A in the human pre-C* structure (*Zhan et al., 2022*). However, the resolutions of the yeast C* and P complex structures make unambiguous assignment of Fyv6 amino acid side chains and interactions difficult.

Here, we use a combination of transcriptomic, structural, biochemical, and genetic assays to elucidate how Fyv6 controls 3′ SS usage in yeast. RNA-seq analysis reveals widespread activation of alternative, BP proximal non-consensus 3′ SS in the absence of Fyv6 consistent with biochemical assays that show Fyv6 facilitates splicing of 3′ SS located ≥21 nt from the BP. To further elucidate Fyv6 function, we determined a 2.3 Å cryo-EM structure of the yeast P complex spliceosome in which Fyv6 can be modeled. The structure reveals interactions between Fyv6 and Prp22 as well as a domain in Syf1 that interacts with either Fyv6 or the first step factor Yju2. 3D classification allowed us to identify two additional conformational states of P complex and provides insights into the coupling of 3′ SS active site docking and the presence of second step factors. We then used this structural data to dissect the functional domains of Fyv6, interpret the results of a genetic screen for suppressors of *fyv6Δ*, and to probe Fyv6 and Prp22 interactions. Combined, we propose a model in which Fyv6 is recruited to the spliceosome via interactions with the NTC component Syf1 in order to promote usage of BP distal, and Prp22-dependent, 3′ SS.

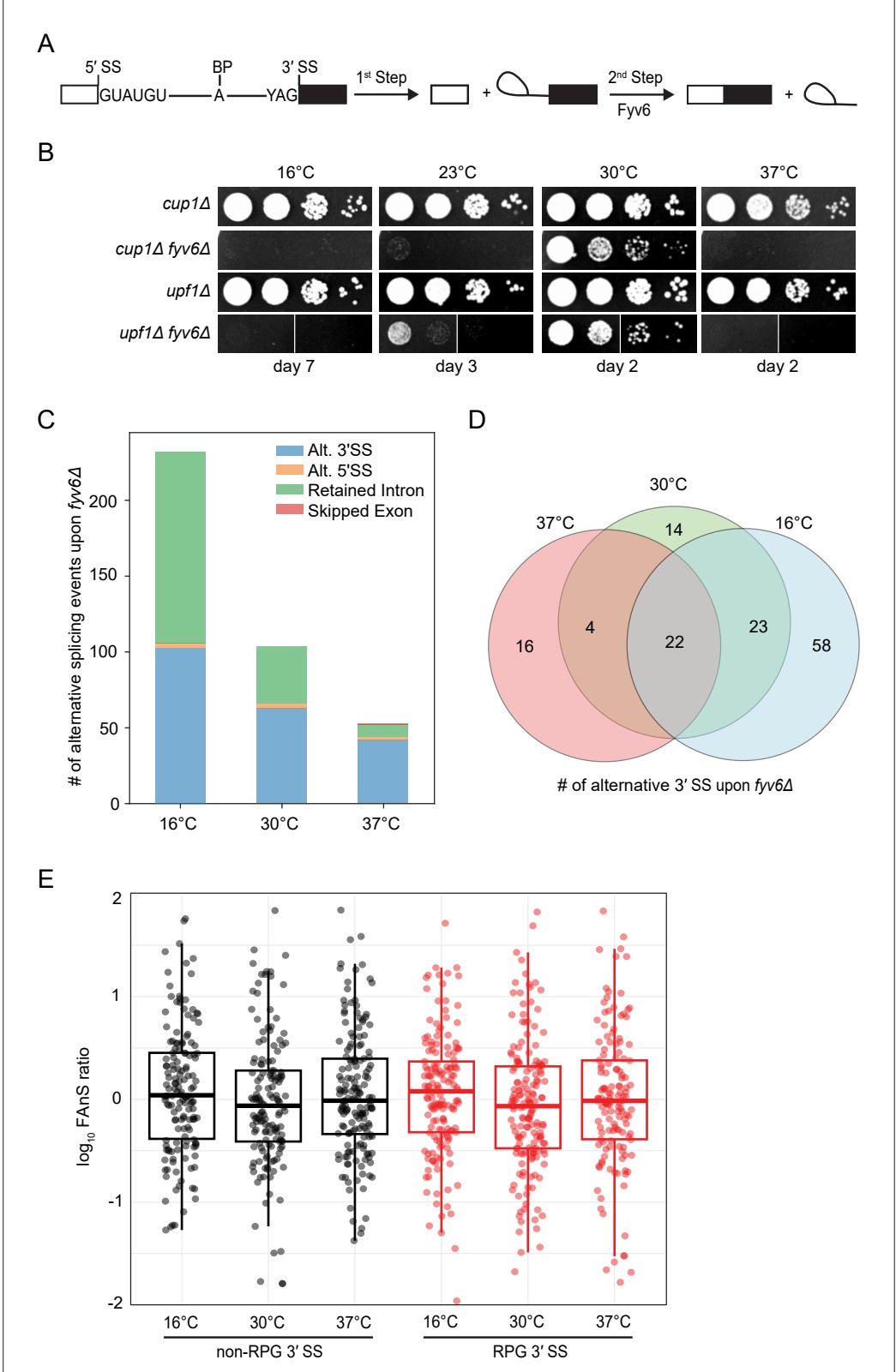

**Figure 1.** Widespread splicing changes due to loss of Fyv6. (**A**) Precursor messenger RNA (pre-mRNA) splicing reaction steps and products. (**B**) Growth of WT (*cup1Δ*), *fyv6Δ*, *upf1Δ*, and *fyv6Δupf1Δ* yeast at 16°C, 23°C, 30°C, and 37°C. Images were taken on the indicated days. White lines indicate where images were cropped, as all dilutions for all strains were on the same plate, but not necessarily directly adjacent to one another. (**C**) Number of

*Figure 1 continued on next page*

*Figure 1 continued*

alternative splicing events observed in a *fyv6Δupf1Δ* strain relative to a strain containing FYV6 after a temperature shift to 16°C, 30°C, and 37°C. (**D**) Venn diagram of shared alternative 3′ splice site (SS) splicing events at 16°C (blue), 30°C (green), and 37°C (red). (**E**) Box plots of the $\log_{10}$ of the ratios of *fyv6Δ upf1Δ* to WT (*upf1Δ*) fraction of annotated splicing (FAnS) values for ribosomal protein genes (RPGs) (red) and non-RPGs (black) for 3′ SS splicing events. FAnS values represent the ratio of the alternative splicing event (here, selection of an alternate 3′ SS) to the annotated, canonical splice isoform.

The online version of this article includes the following source data and figure supplement(s) for figure 1:

**Figure supplement 1.** Gene expression analysis based on the RNA sequencing (RNA-seq) results and an example of Fyv6-dependent splicing changes in *YOS1*.

**Figure supplement 1—source data 1.** TIF file containing original gel image for *Figure 1—figure supplement 1F*, indicating the relevant bands.

**Figure supplement 1—source data 2.** The original gel image for *Figure 1—figure supplement 1F*.

## Results

### Deletion of *FYV6* results in widespread use of alternative 3′ SS

Our previous work showed that loss of *FYV6* causes changes in 3′ SS usage in the *SUS1* pre-mRNA. To probe for changes transcriptome-wide, we used nonsense mediated decay (NMD)-deficient (*upf1Δ*, WT) yeast cells expressing or lacking Fyv6. By suppressing NMD, we hoped to limit degradation of alternatively spliced mRNA isoforms generated in the absence of Fyv6 (*Sayani et al., 2008*). Consistent with results from *fyv6Δ* strains, the *fyv6Δ upf1Δ* double mutant strain also showed reduced growth at 30°C and cold-sensitive (*cs*) and temperature-sensitive (*ts*) phenotypes relative to a *upf1Δ* control strain (*Figure 1B*). We then used RNA-seq to analyze the isolated RNAs from these strains. As expected, many more changes in splicing were observed in the *fyv6Δ upf1Δ* strain relative to the *fyv6Δ* strain (*Figure 1—figure supplement 1A and B*; *Supplementary file 1*, *Supplementary file 2*). This is consistent with many mRNA isoforms generated due to loss of Fyv6 being substrates for NMD. Loss of Fyv6 resulted in a number of changes in gene expression with many non-intron-containing genes also being up- or downregulated (*Figure 1—figure supplement 1C*).

We compared and classified changes in splicing between strains containing or lacking *FYV6* under permissive growth conditions (30°C) or after the yeast had been shifted to a non-permissive temperature (16°C or 37°C) for 1 hr (*Figure 1C*). In this case, we only considered events resulting in at least a 10% change in the percent spliced in value when *FYV6* is deleted. At all temperatures, we see changes in alternative 3′ SS usage in the *fyv6Δ* strain relative to the *FYV6* background. For example, we detected use of the alternative 3′ SS in the first intron of the *SUS1* transcript as we previously reported (*Figure 1—figure supplement 1D*; *Lipinski et al., 2023*). We also saw increased use of a cryptic, non-consensus GAG 3′ SS in the first, but not second, intron of the *YOS1* transcript in the *fyv6Δ* datasets and confirmed this result by RT-PCR (*Figure 1—figure supplement 1E and F*). We observed several cases of alternative 5′ SS. However, the majority of these alternative 5′ SS were used with equal efficiencies in the WT and *fyv6Δ* strains (*Figure 1—figure supplement 1G*).

The strains temperature shifted to 16°C had the largest number of changes due to *FYV6* deletion and the strains shifted to 37°C the least. The temperature shift to 16°C caused an increase in the numbers of both alternative 3′ SS and retained intron events detected (*Figure 1C*). The changes in alternative 3′ SS used at each temperature were mostly unique with few events detected under all three conditions (*Figure 1D*). Collectively, we observed that 61 different RNAs (~20% of introns) changed splicing patterns due to loss of Fyv6. For many of these mRNAs, we detected multiple different 3′ SS being used.

To investigate these splicing events more closely, we analyzed changes in usage of alternative splice junctions by calculating the fraction of annotated splicing (FAnS) in the presence and absence of Fyv6 (*Roy et al., 2023*). FAnS ratios report the relative abundance of an alternative splicing event in relation to the main spliced isoform. We calculated FAnS values for all exon junction reads corresponding to usage of a canonical 5′ SS and an alternative 3′ SS for WT and *fyv6Δ* strains at each temperature. We then calculated the ratios of the *fyv6Δ*/WT FAnS values for each transcript. These results confirm changes in alternative 3′ SS usage for both ribosomal protein gene (RPG) and non-RPG transcripts at all conditions (*Figure 1E*). We could not detect any significant differences between RPG

and non-RPG splicing with this analysis, suggesting that differences in gene expression and splicing efficiencies between these two classes are not correlated with alternative 3′ SS usage. Combined these results show that Fyv6 plays a critical role in defining mRNA isoform production in yeast, particularly under non-optimal growth conditions such as temperature stress.

## Fyv6 facilitates usage of BP distal 3′ SS

We next analyzed the features of the 3′ SS impacted by Fyv6 loss. We used RNAs collected from yeast grown at 30°C and sequenced the libraries at higher depth than those used for the temperature-shift experiment described in *Figure 1* (~400 million vs. 200 million reads; *Supplementary file 2*). We first identified alternative 3′ SS used in each strain, annotated each as being upstream (5′) or downstream (3′) of the canonical 3′ SS (*Grate and Ares, 2002*), and then calculated the FAnS ratio for each site. This analysis shows that the majority of the alternative 3′ SS arising from Fyv6 deletion are upstream (*Figure 2A*, pink points above the diagonal). When these alternative 3′ SS are mapped according to their nucleotide distance from the canonical site, most of these are found within ~40 nt upstream (*Figure 2B*, purple violin). Plotting these alternative 3′ SS sites relative to their distance from the annotated BP reveals that the highest density of sites is ≤20 nt from the BP for transcripts with the highest FAnS ratios (*Figure 2C*, purple violin). It is important to note that we did not map BP in Fyv6 deletion strains, and it is possible that in some cases a shift in 3′ SS usage could also coincide with a shift in the BP. Indeed, a small number of alternative 3′ SS detected in the Fyv6 datasets are located upstream of the annotated BP, suggesting activation of an alternative BP and 3′ SS. However, we believe that the changes are predominantly due to usage of alternative 3′ SS located between the annotated BP and canonical 3′ SS.

The above results suggest that Fyv6 is important for splicing of 3′ SS located distant from the BP. To test this directly and systematically, we created a series of ACT1-CUP1 reporters with various BP to 3′ SS distances (9–50 nt, *Figure 2D*, *Supplementary file 3*) based on sequences used in previous studies of 3′ SS selection (*Brys and Schwer, 1996*; *Frank and Guthrie, 1992*; *Schwer and Gross, 1998*). We transformed strains containing or lacking Fyv6 and without debranchase (Dbr1) to limit degradation of lariat intermediates. We then detected RNA products by primer extension and calculated the second step (exon ligation) efficiency for each reporter (*Figure 2E and F*). While loss of Fyv6 has minimal impact on exon ligation when BP to 3′ SS distances are short (9–15 nt), exon ligation efficiency decreases significantly at distances of 21 nt or greater. In addition, we do not see evidence of cryptic BP usage in these assays. This result is consistent with the RNA-seq analysis and supports a function for Fyv6 in facilitating splicing at BP distal 3′ SS.

Finally, we analyzed the sequence features of the alternative 3′ SS with increased use after loss of Fyv6 (*Figure 2G and H*). For sites located upstream of the canonical 3′ SS, we observed much greater sequence variability relative to the canonical site. We detected increased use of highly variable 3′ SS including those with atypical HAU (H=A,C,U) and BG (B=C,U,G) motifs and confirmed generation of new RNA isoforms for several of these by RT-PCR (*Figure 2H*, *Figure 2—figure supplement 1*). Increased use of variable 3′ SS was also observed when the second step factor *PRP18* was deleted (*Roy et al., 2023*). However, the subset of RNAs most impacted by splicing factor deletion appear to be different. For example, we were not able to detect changes in 3′ SS usage for the *UBC12*, *MAF1*, *MUD1*, *PHO85*, *SPT14*, or *YCL002C* transcripts in the *fyv6Δ* datasets as was reported for *prp18Δ* (not shown). Prp18 and Fyv6 likely have transcript-specific effects on splicing outcomes. Combined our results strongly support Fyv6 as a second step splicing factor that facilitates usage of consensus sites located distal to the BP in vivo.

## A high-resolution spliceosome structure reveals Fyv6 interactions

Previous structures of the yeast P complex spliceosome were solved with resolutions at the core ranging from 3.3 Å to 3.7 Å (*Bai et al., 2017*; *Liu et al., 2017*; *Wilkinson et al., 2017*), and with peripheral components such as the Prp22 helicase, U2 snRNP, and Prp19 complex (NTC) at much lower resolutions (5–10 Å) that precluded detailed investigation. To improve the resolution, we purified P complex as previously described by stalling spliceosome disassembly with a dominant negative mutant of Prp22 (S635A) defective in mRNA release (*Schwer and Meszaros, 2000*; *Wilkinson et al., 2017*). However, by collecting a much larger cryo-EM dataset on a more modern electron detector, we were able to resolve the structure to 2.3 Å within the catalytic core, as well as visualize peripheral

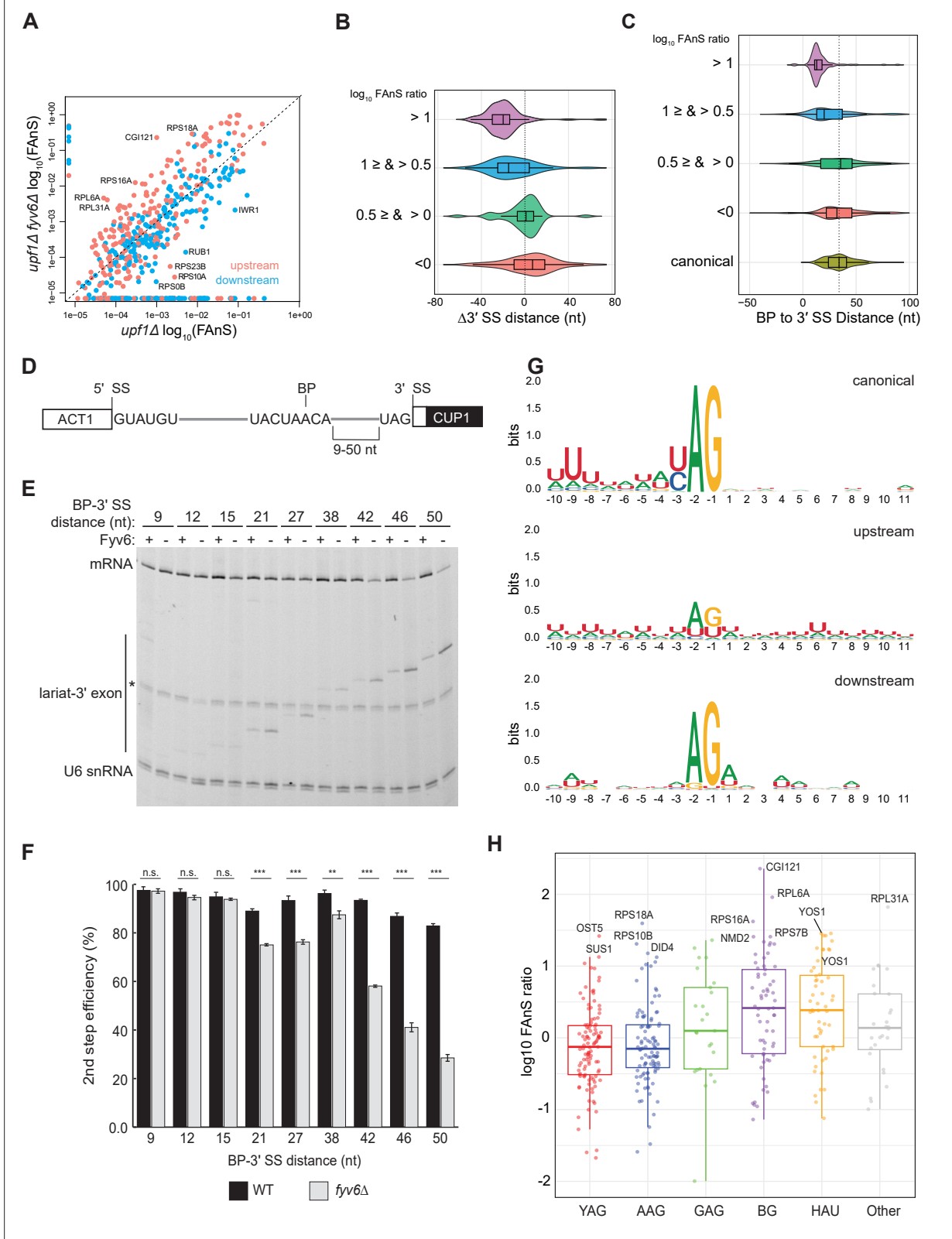

**Figure 2.** Loss of Fyv6 activates branch point (BP) proximal, non-consensus 3′ splice site (SS). (**A**) Plot of the $\log_{10}$ of fraction of annotated splicing (FAnS) values for differences between alternative 3′ SS usage in *fyv6Δ upf1Δ* and *upf1Δ* strains. High FAnS values indicate increased usage of the alternate 3′ SS relative to the annotated site. Points are colored based on whether they are upstream (pink) or downstream (blue) to the canonical 3′ SS. (**B**) Violin plots of the ratios of *fyv6Δupf1Δ* to the *upf1Δ* FAnS values based on the distances between the alternative and canonical 3′ SS ($\Delta$3′ SS = 3′ $SS_{Alt.}$ − 3′

*Figure 2 continued on next page*

*Figure 2 continued*

SS$_{canonical}$). FAnS ratios>0 are indicative of the site being upregulated in *fyv6Δ*. (**C**) Violin plots of the ratios of *fyv6Δupf1Δ* to the *upf1Δ* FAnS values based on the distances between the alternative 3′ SS and the annotated BP. FAnS ratios >0 are indicative of the site being upregulated in *fyv6Δ*. (**D**) Diagram of ACT1-CUP1 reporter showing BP-3′ SS distances. (**E**) Representative primer extension analysis of RNA products generated from splicing of the ACT1-CUP1 reporter in the presence (WT) or absence of Fyv6 (*fyv6Δ*). Bands for fully spliced mRNA and lariat intermediate are indicated. U6 small nuclear RNA (snRNA) was detected as a loading control. The * indicates an unknown product present in every lane. (**F**) Quantification of the primer extension results from N=3 biological replicates represented by the ratio of band intensities for mRNA/(mRNA+lariat intermediate). Bars represent the average ratio of the replicates ± SD. Means between WT and *fyv6Δ* groups for each reporter were compared with an unpaired Welch's two-tailed t-test. Significance is indicated: n.s., no significance; **p<0.01; ***p<0.001. (**G**) Sequence logos of alternative 3′ SS with FAnS>0 sorted by either upstream or downstream of the canonical 3′ SS compared against the canonical 3′ SS for genes with FAnS>0. (**H**) The log$_{10}$ of the FAnS ratio for alternative 3′ SS sorted by the sequences of the 3′ SS.

The online version of this article includes the following source data and figure supplement(s) for figure 2:

**Source data 1.** TIF file containing original gel image for *Figure 2E*, indicating the relevant bands.

**Source data 2.** The original gel image for *Figure 2E*.

**Figure supplement 1.** Examples of Fyv6-dependent alternative 3′ splice site (SS) usage.

**Figure supplement 1—source data 1.** TIF files containing original gel images for *Figure 2—figure supplement 1*, indicating the relevant bands.

**Figure supplement 1—source data 2.** The original gel images for *Figure 2—figure supplement 1*.

components at resolutions from 3.0 Å to 3.7 Å (*Figure 3A and B*; *Figure 3—figure supplements 1–4*; *Supplementary file 4*). This is the highest resolution spliceosome structure to date. The high-quality density at the active site shows the positions of bases unambiguously, confirming the manner of the non-Watson-Crick base pairing that mediates 3′ SS recognition (*Liu et al., 2017*; *Wilkinson et al., 2017*). Monovalent ions bound at the active site were previously shown to be important during the first step of splicing (*Wilkinson et al., 2021*), and our data indicate that the positions of these ions are preserved after exon ligation. Additionally, the density also shows a metal ion, likely potassium, that bridges the base pair between +1G of the 5′ SS and −1G of the 3′ SS (*Figure 3C*).

The second step splicing factor Prp22 is resolved at 3.0 Å resolution, allowing atomic interpretation for the first time (*Figure 3B* and later figures). Bases 13–21 of the 3′ exon are visible between the two RecA domains of Prp22 consistent with biochemical footprinting data (*Schwer, 2008*; *Figure 3—figure supplement 4*). The S635A mutation used to stall P complex is also visible; however, the origin of defective mRNA release due to this mutation is not apparent (*Schwer and Meszaros, 2000*). The loop containing S635A is in a similar conformation in this structure and in the structure of WT human Prp22 (*Felisberto-Rodrigues et al., 2019*; *Figure 3—figure supplement 4*). Part of the extensive N-terminal domain of Prp22 is resolved and constitutes a long helix that bridges between the Prp22 RecA2 domain and the C-terminal domain of Cwc22; this helix was unassigned in previous P complex structures (*Figure 4B*).

To improve the resolution of the peripheral, flexible regions, we used a new data-driven regularization strategy (*Kimanius et al., 2024*) that allows focused refinements of much smaller domains than previously possible (*Figure 3—figure supplements 2 and 4*). Combined with AlphaFold2-assisted modeling, we were able to obtain more accurate models for the U2 snRNP, NTC, U5 Sm ring, and Cwc22 N-terminal domains. Our model for the NTC within the P complex is similar to our previous model for the NTC within C and C$^i$ complexes (*Wilkinson et al., 2021*; *Figure 4C*). Notably however, we observed a star-shaped density coordinated by several conserved lysines from Clf1 and Ntc20, which we were able to model as inositol hexakisphosphate (IP$_6$) (*Figure 4D*). The same density, at lower resolution, is visible in the C complex (*Wilkinson et al., 2021*). A distinct, separate IP$_6$ molecule had previously been observed within all catalytic spliceosomes between the B$^{act}$ and P complex stages coordinated by Prp8 and (for C* and P complexes) Slu7 (*Figure 3—figure supplement 4*). We do not believe that the NTC-bound IP$_6$ is essential for splicing since deletion of the Clf1 tetratricopeptide repeat that coordinates IP$_6$ does not result in growth or splicing phenotypes in yeast and Ntc20 is a nonessential splicing factor (*Chen et al., 2001*; *Chung et al., 1999*). Nonetheless, it is possible that IP$_6$ may regulate splicing in some manner via interactions with both Prp8 and the NTC.

We were able to unambiguously assign three, connected long α-helices to Fyv6 (*Figure 4A*). These helices were visible in previous cryo-EM reconstructions of yeast C* and P complexes but were either unassigned or misassigned. In the cases of misassigned densities, Fyv6 was previously attributed to the C-terminal domain of the first step factor Yju2 (*Wilkinson et al., 2017*). Yju2 also has an elongated

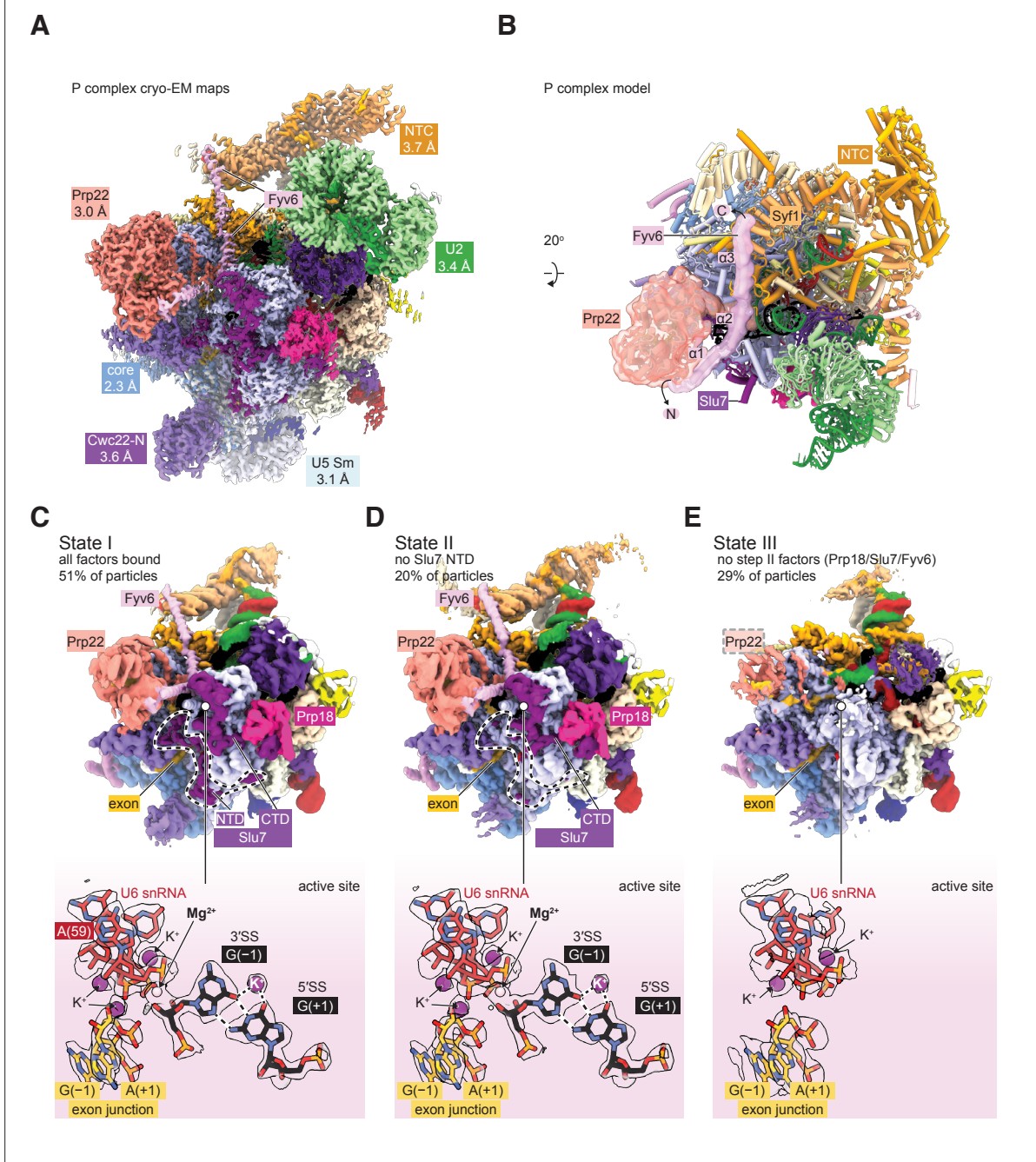

**Figure 3.** Cryo-electron microscopy (cryo-EM) structure of the yeast P complex spliceosome at 2.3 Å resolution. (**A**) A composite density map for P complex showing focused refinements of the Prp22, NTC, U2 small nuclear ribonucleoprotein (snRNP), U5 Sm ring, and Cwc22 N-terminal domain regions. (**B**) Overall model for the P complex spliceosome. (**C**) Cryo-EM density for state I of P complex (above, low-pass filtered) and for the active site and 3′ SS (below, sharpened). (**D, E**) As for (C) but for states II (**D**) and III (**E**).

The online version of this article includes the following figure supplement(s) for figure 3:

**Figure supplement 1.** Spliceosome cryo-electron microscopy (cryo-EM) data collection and general processing.

**Figure supplement 2.** Focused classification and refinement scheme for regions of P complex.

**Figure supplement 3.** Properties of the cryo-electron microscopy (cryo-EM) datasets and models.

**Figure supplement 4.** Features of the cryo-electron microscopy (cryo-EM) model of P complex.

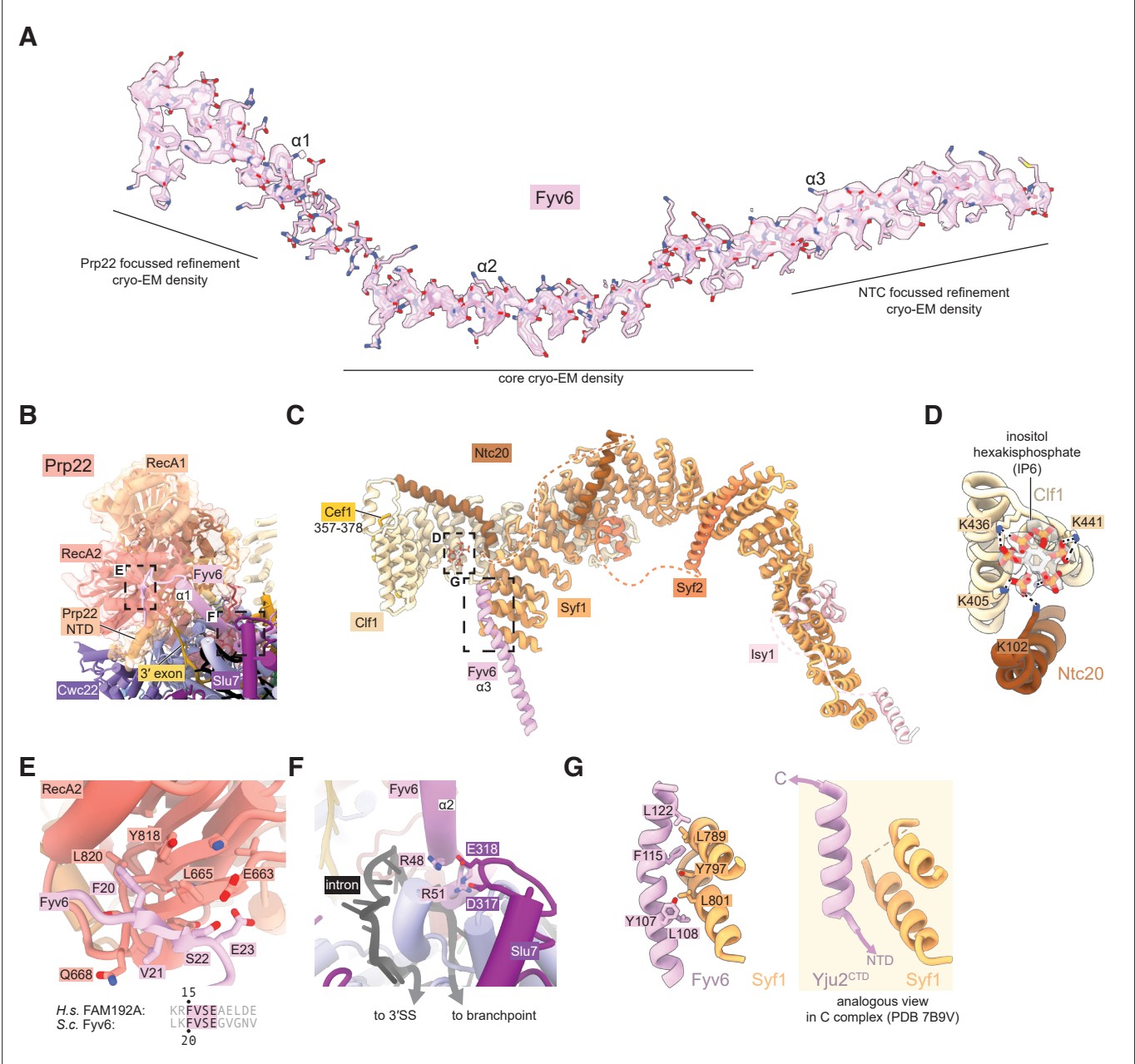

**Figure 4.** Fyv6 and its interactors in the P complex cryo-electron microscopy (cryo-EM) structure. (**A**) Cryo-EM density segmented around Fyv6. (**B**) Structure of Prp22. (**C**) Structure of the NTC within P complex. (**D**) IP₆ site and cryo-EM density in NTC. (**E**) Interaction of the hook domain of Fyv6 with the Prp22 RecA2 domain. (**F**) Interaction of Fyv6 with Slu7 and the region of the intron between the branch point (BP) and 3′ splice site (SS). (**G**) Interaction of Fyv6 with Syf1 and an analogous view of the interaction of Syf1 with Yju2 in C complex (*Wilkinson et al., 2021*).

The online version of this article includes the following figure supplement(s) for figure 4:

**Figure supplement 1.** Comparison of fits to density of Yju2 vs. Fyv6 in each complex.

**Figure supplement 2.** Alignments of Fyv6 homolog protein sequences.

helical architecture and adopts a similar position on C complex (*Wilkinson et al., 2021*) and the post-P complex intron lariat spliceosome (ILS) (*Wan et al., 2017*). We closely inspected the density from previously determined spliceosome structures and concluded that the Yju2 C-terminus is still a better fit than Fyv6 for densities in B*, C, and ILS complexes, whereas Fyv6 is a better fit than Yju2 for densities in C* and P complexes (*Figure 4—figure supplement 1*). Therefore, this position of Fyv6

seems to be characteristic of a second step conformation of the spliceosome and consistent with the position of its distant human homolog FAM192A in the human pre-C* complex (*Zhan et al., 2022*).

Fyv6 makes a multitude of interactions with essential splicing factors (*Figures 3B and 4E–G*). One conserved region of Fyv6 is the N-terminus with an FVSE motif (*Figure 4—figure supplement 2*). This motif forms a 'hook' that interacts with a hydrophobic patch on the surface of the Prp22 RecA2 domain, where it also forms hydrogen bonds by β-sheet augmentation (*Figure 4E*). The hook is followed by three α-helices. The start of helix 2 sits on top of the intron between the BP and the active site docked 3′ SS, where it may act as a steric block to prevent 3′ SS undocking and promote exon ligation (*Figure 4F*; *Liu et al., 2017*). Helix 2 also contains a patch of conserved arginines that form salt bridges with Asp317 and Glu318 of Slu7 (*Figure 4F*). The linker between helices 2 and 3 interacts with the NTC component Cef1, and helix 3 interacts with NTC component Syf1 (*Figure 4G*). The binding interface with Syf1 overlaps with the Syf1/Yju2 interaction in the C complex (*Wilkinson et al., 2021*), suggesting these interactions are mutually exclusive (*Figure 4G*). Fyv6 thus provides a direct link between the Prp22 ATPase, NTC components, and other essential splicing factors over a distance of ~100 Å.

By 3D classification, we were able to resolve a total of three states of the P complex spliceosome (*Figure 3C–E*). State I, represented by 51% of particles, is the most complete and has all the above-described factors visible, including a stably docked 3′ SS. State II, represented by 20% of particles, is similar to state I but lacks density for the N-terminal half of the second step factor Slu7. The endonuclease-like domain of Prp8, to which this domain of Slu7 binds, is correspondingly shifted. Interestingly, this state retains strong density for the 3′ SS, suggesting the N-terminal regions of Slu7 are not involved in maintaining 3′ SS docking after exon ligation. Finally, state III, containing 29% of particles, entirely lacks density for the second step factors Slu7, Prp18, and Fyv6. Prp17 and the RNase H domain of Prp8 are still present but are weaker than in States I and II, suggesting they are more flexible in state III. Most of Prp22 can still be observed with weaker density except for the C-terminal domain that interacts with Prp8, which lacks density entirely. Importantly, state III also lacks density for the 3′ SS but has density for the 3′ exon, suggesting it is indeed post-catalytic and represents a state after loss of second step factors and undocking of the 3′ SS from the catalytic core. The coincidence of loss of Fyv6 with loss of Prp18 and Slu7 reinforces the notion that these three factors act together and are important for 3′ SS docking, maintaining the 3′ SS in the active site after exon ligation, and for stabilization of Prp17, Prp22, and the RNase H domain of Prp8.

## The Fyv6/Syf1 interaction is critical for suppressing *fyv6Δ* phenotypes

To determine the functionally critical regions and interactions of Fyv6, we used the cryo-EM structure to design five truncation mutants (*Figure 5A*). The first three truncations remove amino acids from the N-terminus of Fyv6: deletion of the first 16 amino acids, which are not resolved in the cryo-EM structure (Δ1–16); deletion of the first 23 amino acids, which removes the conserved hook region of Fyv6 that interacts with the Prp22 RecA2 domain (Δ1–23); and deletion of amino acids 1–51, which additionally removes the first α-helix and interactions with Prp22, Prp8, and Slu7 (Δ1–51). The remaining two mutants are C-terminal truncations. In the first, amino acids 134–173, which do not have clear density in the cryo-EM structure and are not modeled but which do contain a predicted nuclear localization signal (NLS), are deleted (Δ134–173) (*Nguyen Ba et al., 2009*). In the second, amino acids 103–173 are deleted (Δ103–173), which additionally removes the region of the protein that interacts with Syf1.

We tested each truncation mutant for its ability to suppress the *cs* and *ts* phenotypes of the *fyv6Δ* strain when expressed from a plasmid under the native promoter (*Figure 5B*). All of the N-terminal truncation mutations could suppress the *cs* or *ts* phenotypes to some degree. Surprisingly, a strain with the complete deletion of the conserved hook domain that interacts with Prp22 (Δ1–23) grew similarly to the strain expressing WT Fyv6. We wondered if splicing changes could still be present even if the growth phenotype was suppressed; so, we assayed *SUS1* isoforms by RT-PCR. We previously showed that loss of Fyv6 causes use of an alternative 3′ SS in the first intron of this pre-mRNA (*Lipinski et al., 2023*). In addition to suppression of the *fyv6Δ* growth phenotype, the N-terminal truncation mutants also showed no evidence for changes in *SUS1* splicing (*Figure 5C*). The N-terminal region of Fyv6 composed of the hook domain and the first α-helix is not required for suppression of either the

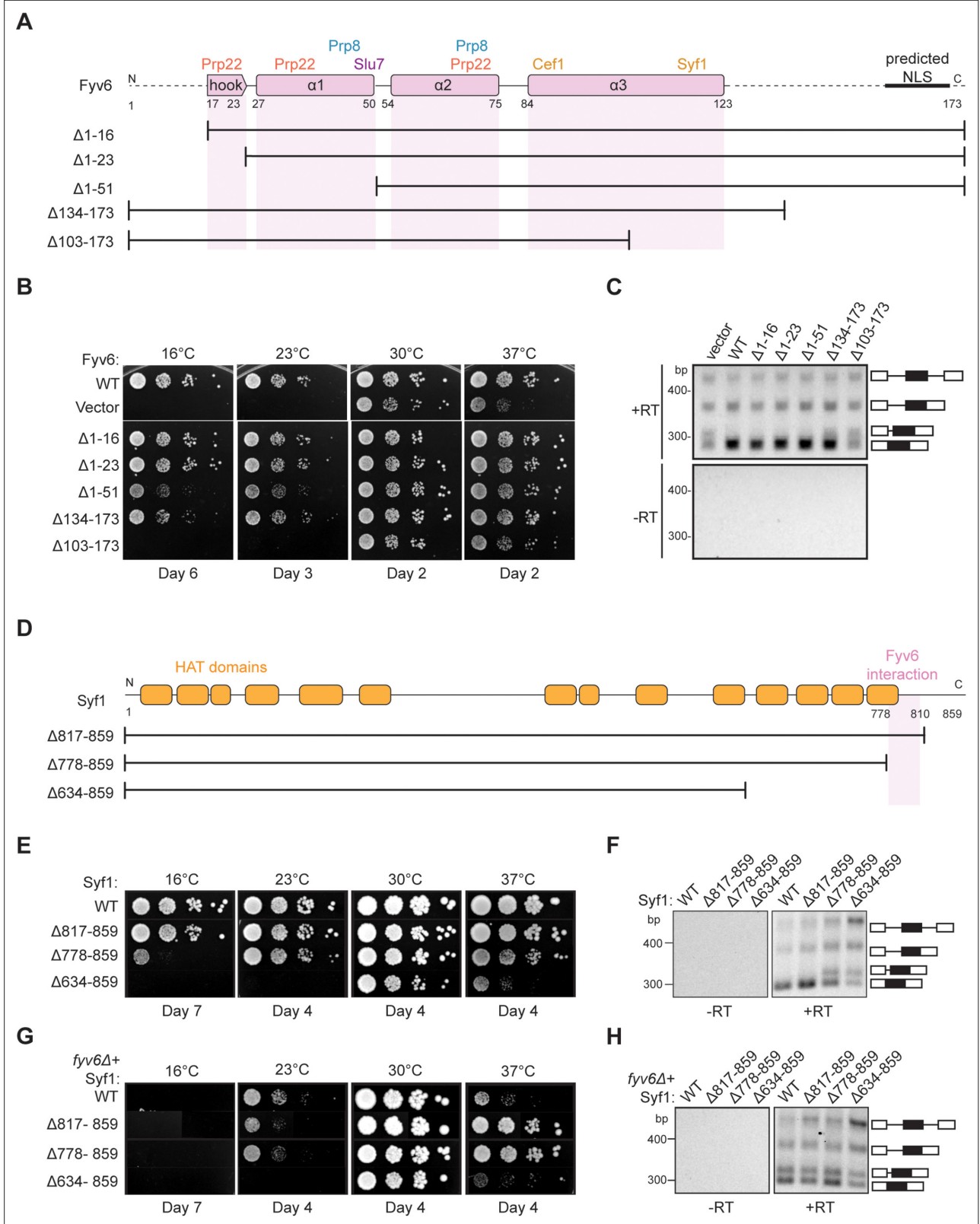

**Figure 5.** Structure-based analysis of Fyv6 domain function. (**A**) Diagram of Fyv6 protein structure, protein interactors, and protein truncations. NLS = nuclear localization signal. (**B**) Spot dilution assay of strains with Fyv6 truncations on –Trp DO plates at different temperatures. Plates were imaged after the number of days indicated. (**C**) Representative gel image of *SUS1* RT-PCR in strains with an empty vector or expressing the indicated Fyv6 variant. (+RT reactions contain reverse transcriptase; –RT control reactions do not contain reverse transcriptase.) (**D**) Diagram of Syf1 protein structure and

*Figure 5 continued on next page*

*Figure 5 continued*

truncations. (**E**) Spot dilution assay of strains with Syf1 truncations on –Trp DO plates at different temperatures. Plates were imaged after the number of days indicated. (**F**) Representative gel image of *SUS1* RT-PCR in strains with an empty vector or expressing the indicated Syf1 variant. (**G**) Temperature growth assay of strains with Syf1 truncations in a *fyv6Δ* background on YPD plates. Plates were imaged after the number of days indicated. (**H**) Representative gel image of RT-PCR of *SUS1* in strains with plasmids containing WT Syf1 or a Syf1 truncation in a *fyv6Δ* background.

The online version of this article includes the following source data and figure supplement(s) for figure 5:

**Source data 1.** TIF files containing original blot images for *Figure 5C, F, and H*, indicating the relevant bands.

**Source data 2.** The original gel images for *Figure 5C, F, and H*.

**Figure supplement 1.** Expression of tagged Fyv6 constructs.

**Figure supplement 1—source data 1.** TIF files containing original blot images for *Figure 5—figure supplement 1A*, indicating the relevant bands.

**Figure supplement 1—source data 2.** The original images for *Figure 5—figure supplement 1A*.

temperature sensitivity or splicing phenotypes observed when Fyv6 is lost despite its conservation and interactions with critical splicing factors (Prp8, Prp22, and Slu7).

For the C-terminal truncations, the Δ134–173 truncation was also able to suppress the *cs* and *ts* phenotypes, indicating that the predicted NLS is also not essential (*Figure 5B*). However, this mutant grew more poorly at 16°C than did the Δ1–16 and Δ1–23 truncations and showed some evidence of alternative 3′ SS usage in *SUS1* (*Figure 5C*). The largest effects were seen when the C-terminus was truncated further. The Δ103–173 truncation strain phenocopied the *fyv6Δ* strain and grew poorly at 37°C and was dead at 23°C or 16°C (*Figure 5B*). Additionally, RT-PCR showed increased use of the alternative 3′ SS in *SUS1* with this truncation (*Figure 5C*). To determine if this phenotype could be due to loss of protein expression, we assayed expression of the epitope-tagged proteins by western blot (*Figure 5—figure supplement 1A*). In all cases, we could detect protein; however, the abundance of the N-terminally epitope-tagged Δ103–173 variant was much lower than the others. Therefore, we also constructed a C-terminally epitope-tagged version of Δ103–173 (*Figure 5—figure supplement 1B*). The C-terminally tagged protein expressed at much higher levels but also failed to suppress the *cs* and *ts* phenotypes. These data suggest that the C-terminal region of Fyv6 that includes a portion of α-helix 3 and the Syf1 interaction domain is critical for function.

To obtain additional evidence for the importance of the Fyv6/Syf1 interaction, we created a *syf1Δ* shuffle strain containing WT Fyv6 and expressed C-terminal Syf1 truncations (*Figure 5D*). We hypothesized that loss of the Fyv6-interacting domain on Syf1 would phenocopy *fyv6Δ*. We observed just this effect. Loss of the Fyv6-interacting region in Syf1 Δ778–859 and Δ634–859 mutants resulted in either modest (Δ778–859) or severe (Δ634–859) *cs* and *ts* phenotypes. In addition, these Syf1 truncation mutants also triggered use of the alternative 3′ SS in *SUS1* (*Figure 5F*). Interestingly, if the Δ778–859 and Δ817–859 Syf1 mutants are expressed in a *fyv6Δ* background, it results in suppression of the *fyv6Δ ts* phenotype (*Figure 5G*; *Figure 5B*). However, the *SUS1* alternative 3′ SS is still used (*Figure 5H*). This indicates that *ts* phenotype suppression and triggering usage of the alternative 3′ SS in *SUS1* are separable processes.

When considered together, these results likely originate from mutually exclusive Fyv6/Syf1 and Yju2/Syf1 interactions. It is possible that in the absence of Fyv6, Syf1 inappropriately binds Yju2 during exon ligation. The presence of Yju2 during the second step could then lead to growth defects that are suppressed when the Yju2-binding site on Syf1 is deleted. Alternatively or in addition, the absence of Fyv6 could perturb the equilibrium between the first and second step conformations of the spliceosome and result in favoring of the first step (*Liu et al., 2007b*). Loss of this equilibrium could lead to growth defects that are suppressed by destabilization of the first step conformation by removing a binding site for Yju2 on Syf1 and restoring the equilibrium. In both cases, truncation of Syf1 suppresses growth defects but not *SUS1* splicing defects since Fyv6 function is still needed during the second step.

## Mutations in many different splicing factors can suppress *fyv6Δ* phenotypes

To gain additional insights into Fyv6 function, we carried out a genetic screen to detect spontaneously arising suppressors that can correct temperature-dependent growth defects (*Figure 6A*). This method has previously been used to illuminate many aspects of spliceosome biochemistry and non-silent

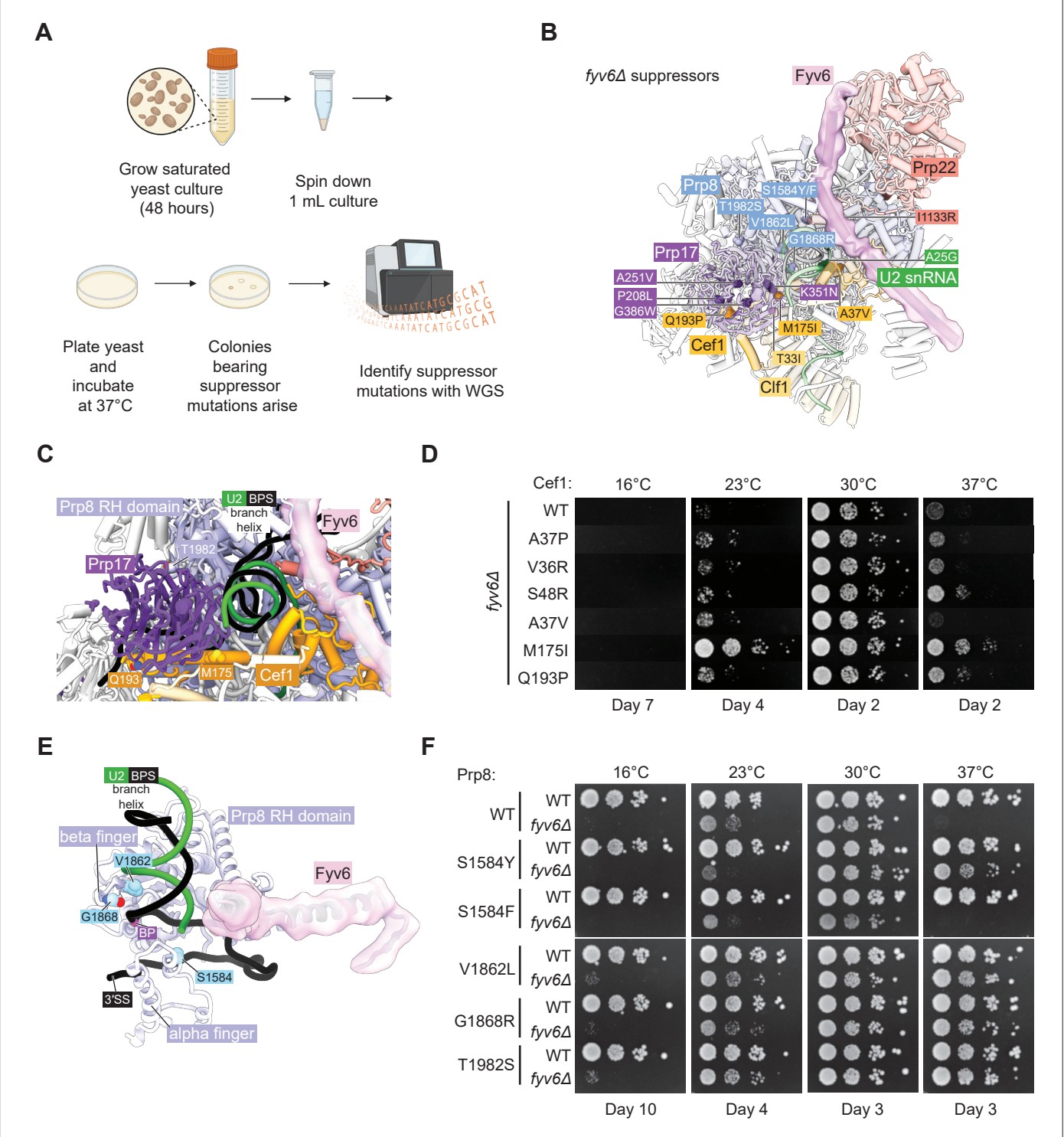

**Figure 6.** Identification of novel suppressors of *fyv6Δ* temperature sensitivity. (**A**) Workflow for the suppressor screen. (**B**) P complex spliceosome structure with *fyv6Δ* suppressor mutations labeled (see **Supplementary file 5**). (**C**) Close-up view of Cef1 and Prp8 suppressor mutations in P complex. (**D**) Spot dilution temperature growth assay with Cef1 mutants on YPD plates. Plates were imaged after the number of days indicated. (**E**) Close-up view of Prp8 mutations in P complex. (**F**) Spot dilution temperature growth assay with Prp8 mutants on YPD plates. Plates were imaged after the number of days indicated.

The online version of this article includes the following figure supplement(s) for figure 6:

**Figure supplement 1.** Isolated suppressor strains for *fyv6Δ* grown at different temperatures.

**Figure supplement 2.** Genetic interactions between Prp18 and Fyv6.

mutations within splicing factors are likely causative for suppression (*Montemayor et al., 2014*; *Kuhn and Brow, 2000*; *Brow, 2019*).

We plated saturated *fyv6Δ* yeast cultures onto 20 different YPD plates and incubated them at 37°C until colonies arose. Within several days, colonies of various sizes appeared and were isolated. A subset of picked colonies were then tested for suppression of the *ts* phenotype. All of these colonies were able to suppress growth defects at 37°C, and many also suppressed the *cs* phenotype at 23°C (*Figure 6—figure supplement 1*). We then performed whole-genome sequencing (WGS) on the largest of the picked colonies as these were expected to be the strongest suppressors. In 19 of 20 sequenced strains, we were able to identify at least one non-silent mutation in a splicing factor (*Figure 6B*, *Supplementary file 5*). When we carried out a similar experiment aimed at identifying *cs* suppressors, we were unable to identify any likely causative mutations in splicing factors. Therefore, we focused our analysis on the *ts* suppressor mutations.

Collectively, the identified mutations tended to fall into two categories: gain of positive charge or substitution with a bulkier side chain. Gain of positive charge mutations were found in Prp8 (G1868R) as well as the second step factors Slu7 (A24R and N28K) and Prp22 (I1133R). Structurally, amino acids within Prp8 and Prp22 are proximal to RNA within the spliceosome and addition of charged amino acid side chains may stabilize interactions with the phosphodiester RNA backbone. The N-terminus of Slu7 containing A24 and N28 is not resolved in any spliceosome structure to our knowledge and normally contains many negatively charged residues. The majority of identified suppressors involve a mutation to a bulkier side chain including several in Prp8 (S1594F/Y, V1862L), Prp17 (A251V, G386W), Cef1 (A37V), and Clf1 (T33I). The increase in hydrophobic surface area provided by these mutations may help to stabilize interactions with nearby proteins, compensating for structural destabilization due to the absence of Fyv6. Alternatively, since Prp8, Prp17, Cef1, and Clf1 are all present during 5'SS cleavage, it is possible that these bulky mutations could destabilize a first step conformation of the spliceosome and suppress phenotypes due to the loss of Fyv6 by facilitating formation of the second step conformation.

We isolated one suppressor mutation within the U2 snRNA (A25G) near where the C-terminal domain of Prp22 inserts toward the active site and located beneath the Fyv6 binding site. It has previously been observed that mutation of this specific U2 snRNA nucleotide is not deleterious to growth (*McPheeters and Abelson, 1992*) and is therefore well tolerated by yeast. The mechanism of suppression by this mutation is not clear; however, it is interesting to note that we also observed suppressor mutations in the U2 snRNP component Rse1 (*Supplementary file 5*) which is released from the spliceosome prior to catalysis. The Rse1 mutations always occurred coincident with mutations in other splicing factors. It is possible that suppressors in factors present prior to or after exon ligation are able to suppress (or enhance suppression of) *fyv6Δ* by changing how pre-mRNAs compete for a limited set of splicing factors or by reducing blocks to pre-spliceosome assembly caused by accumulation of stalled complexes (*Mendoza-Ochoa et al., 2019*; *Munding et al., 2013*). Together these data indicate that mutations in many different splicing factors including second step factors and the U2 snRNA can be isolated from *fyv6Δ* suppressor strains.

## Multiple second step splicing defects can be rescued by the same suppressor

We decided to study several of these mutations in greater detail and to confirm that they were sufficient for *fyv6Δ ts* phenotype suppression. In the case of Cef1, we obtained several mutants located near the interface between Cef1 and the second step factor Prp17: A37V, M175I, and Q193P (*Figure 6C*). This interface has only been observed in structures of C* and P complex spliceosomes, suggesting its importance for exon ligation. In addition, it has previously been reported that another mutant of Cef1 A37 – A37P – is able to suppress splicing defects due to a BS adenine to guanosine substitution (BS-G) which causes stalling after 5' SS cleavage (*Query and Konarska, 2012*). Therefore, we wondered if other Cef1 mutants capable of suppressing defects due to BS-G could also suppress *fyv6Δ* (Cef1 A37P, V36R, and S48R). We created *fyv6Δ* shuffle strains for Cef1 and tested each mutant's ability to suppress *cs* and *ts* phenotypes (*Figure 6D*). All of these Cef1 mutants, both those isolated from our suppressor screen and those identified as BS-G suppressors, could at least weakly suppress the *cs* phenotype at 23°C with our isolated M175I mutant being a particularly strong suppressor. In addition, all of the mutants except Cef1 A37V could suppress the *ts* phenotype at 37°C to varying degrees.

These results confirm that single point mutants in Cef1 can suppress *fyv6Δ* growth phenotypes and that at least some BS-G suppressors in Cef1 also suppress *fyv6Δ*. A common mechanism could be that these mutations stabilize the exon ligation conformation of the spliceosome to favor the second step either when a BS-G is present or Fyv6 is absent.

In the case of Prp8, we tested five different suppressor mutations in *prp8* shuffle strains (*Figure 6E and F*). Two of these are located within the Prp8 α-finger domain (S1584F,Y; *Figure 6E*) while the remainder are located within the RNase H-like domain β-finger (V1862L, G1868R, T1982S; *Figure 6C and E*). We confirmed that four of the identified Prp8 suppressor mutations could partially suppress either or both of the *cs* and *ts fyv6Δ* phenotypes (*Figure 6F*; S1574Y, V1862L, G1868R, and T1982S). Even though we isolated the Prp8 S1584F mutant in our screen and this substitution is very similar to S1584Y, we were not able observe suppression due to this mutant in these strains. Identification of mutants in Prp8 that suppress *fyv6Δ* is not surprising since many different alleles of Prp8 are known to modulate the conformational equilibrium between the first and second steps (*Fica and Nagai, 2017*; *Liu et al., 2007b*; *Query and Konarska, 2004*) or other steps in splicing. Indeed, suppressors in Prp8 have previously been identified at some of these same positions: V1862A/Y/D and T1982A were identified as suppressors of, respectively, the *U4-cs*1 allele (*Kuhn and Brow, 2000*; *Kuhn et al., 1999*) or the 5′ SS U2A mutant (*Siatecka et al., 1999*).

Finally, given our observations with the Cef1 and Prp8 mutants, we wondered about the generality of second step splicing factor suppression. Several dominant mutants in the second step factor Prp18 were previously shown to suppress *ts* phenotypes due to various mutations in Slu7 (*Aronova et al., 2007*). We wondered if these Prp18 mutants would also suppress the *ts* (or *cs*) phenotype due to Fyv6 loss. Indeed, the Prp18 V191A suppressor of *slu7* alleles was able to improve growth at 37°C of *fyv6Δ* yeast (*Figure 6—figure supplement 2*). Together our results from the suppressor analysis support a model in which a given second step suppressor mutation may be able to restore splicing activity lost by a variety of mechanisms.

## A Prp22-dependent splicing stall is relieved by Fyv6 deletion

The final *fyv6Δ* suppressor mutation that we studied was the Prp22 I1133R mutant. This mutation is located within the C-terminal domain of Prp22 that enters the spliceosome active site and is located proximal to U2 snRNA nucleotide A31 (*Figure 7A*). Fyv6 bridges over this Prp22-U2 snRNA interaction. The I1133R mutation might stabilize Prp22/U2 snRNA contacts that can compensate for Fyv6 loss and promote exon ligation. The Prp22 I1133R mutant is able to weakly suppress both the *cs* and *ts* phenotypes in *fyv6Δ* yeast (*Figure 7B and C*). Given its location in the C-terminal domain, we wondered if it or *fyv6Δ* would show synthetic genetic interactions or epistasis with mutations in the helicase domain. We tested two helicase domain mutations, R805A and G810A, both of which are known to impact Prp22-dependent proofreading (*Mayas et al., 2006*). The Prp22 R805A mutant is by itself *cs* (*Schwer and Meszaros, 2000*) and showed a synthetic lethal interaction with *fyv6Δ* (*Figure 7B and C*). This was only slightly rescued by simultaneously introducing I1133R into Prp22. In contrast, Prp22 G810A mutant is viable in the presence and absence of Fyv6 as well as in combination with the I1133R suppressor mutation (*Figure 7B and C*). Prp22 G810A by itself does not suppress the *ts fyv6Δ* phenotype; however, the Prp22 G810A/I1133R double mutant is a stronger suppressor of the *ts* phenotype than either mutant alone. These additive genetic interactions suggest that the Prp22 *fyv6Δ* suppressor mutation (I1133R) and the helicase domain mutations (R805A and G810A) impact distinct Prp22 functions. These functions could correspond to Prp22's role in promoting exon ligation during the second step (the C-terminal domain mutation) and its role in promoting ATP-dependent mRNA release and proofreading (the helicase domain mutations).

Recently, it was reported that the Prp22 G810A mutant could activate usage of some alternative 3′ SS that are also activated by *PRP18* deletion (*Roy et al., 2023*). We therefore wondered if any of the Prp22 mutants could activate alternative 3′ SS also activated by *FYV6* deletion. We isolated RNA from these strains and analyzed *SUS1* splicing by RT-PCR (*Figure 7D and E*). None of the Prp22 mutants activated usage of the *SUS1* alternative 3′ SS or enhanced usage of this site beyond what is observed upon Fyv6 deletion. However, we noticed that the Prp22 helicase mutants, R805A and G810A, both caused an accumulation of unspliced pre-mRNA. This is consistent with these mutations causing a block during pre-mRNA splicing and having a different effect on splicing than Prp22 I1133R. Surprisingly, the block due to the G810A mutation was relieved in the *fyv6Δ* strains and little pre-mRNA

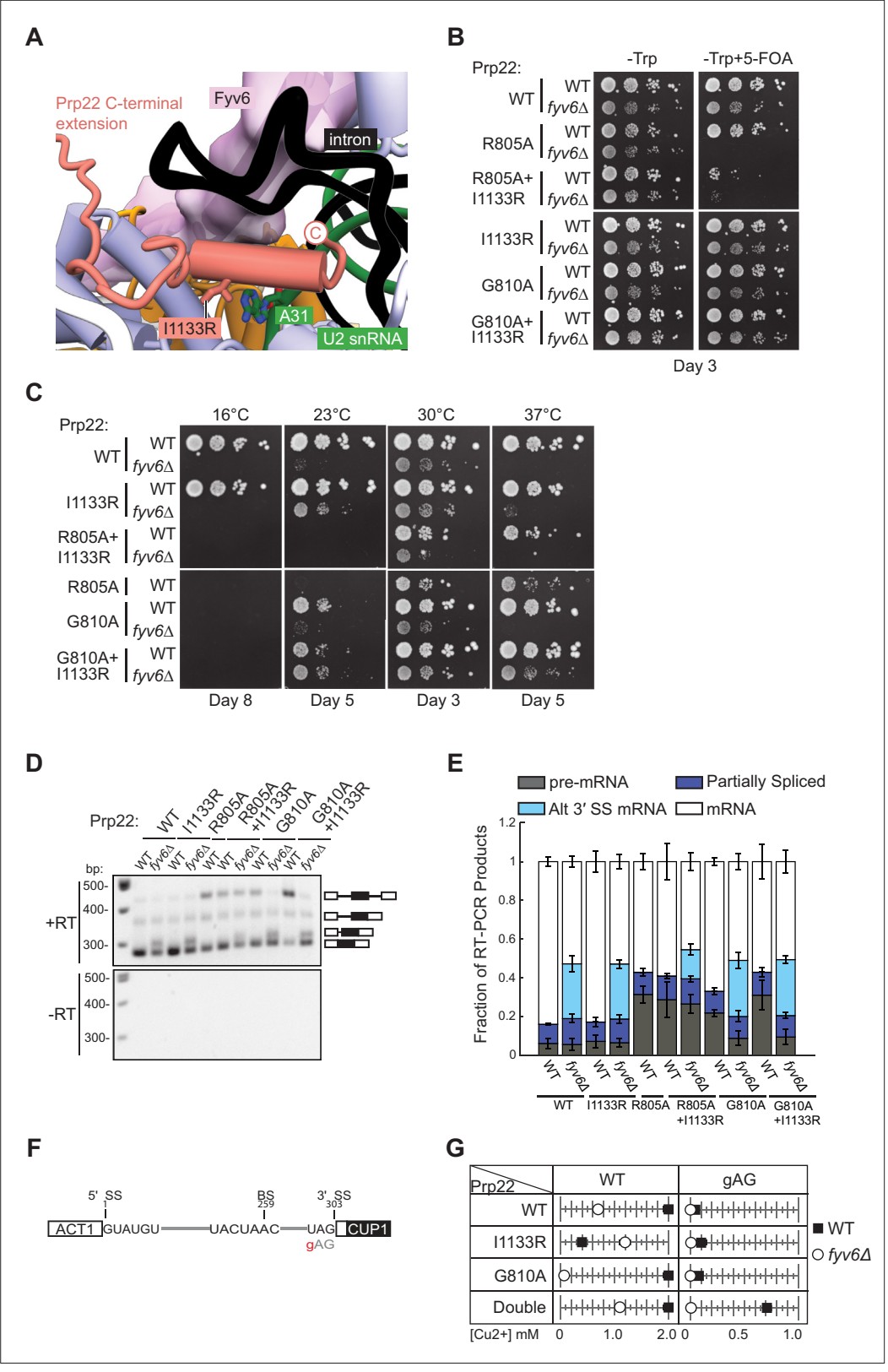

**Figure 7.** Genetic interactions between Prp22 and Fyv6. (**A**) Location of the Prp22 I1133R *fyv6Δ* suppressor in the P complex structure. (**B**) Spot dilution growth assay with Prp22 mutants in WT or *fyv6Δ* backgrounds grown on –Trp DO or –Trp +5 FOA plates. Plates were imaged after 3 days at 30°C. (**C**) Spot dilution growth assay with strains containing Prp22 mutants on YPD plates at different temperatures. Plates were imaged after the number

*Figure 7 continued on next page*

*Figure 7 continued*

of days indicated. (**D**) Representative gel of *SUS1* RT-PCR products in strains with Prp22 mutants in the presence or absence of Fyv6. (**E**) Quantitation of band intensities for each *SUS1* splice isoform as a fraction of total product in the lane. Bars indicate average ± SD of N=3 biological replicates. (**F**) ACT1-CUP1 reporter construct with the location of the 3′ splice site (SS) substitution noted. (**G**) Copper tolerance for each ACT1-CUP1 reporter with each Prp22 allele in strains with (■) or without (O) Fyv6 present. Shown is a representative, single biological replicate of N=3.

The online version of this article includes the following source data and figure supplement(s) for figure 7:

**Source data 1.** TIF files containing original gel images for *Figure 7D*, indicating the relevant bands.

**Source data 2.** The original gel images for *Figure 7D*.

**Figure supplement 1.** ACT1-CUP1 plate images for the data shown in *Figure 7H*.

**Figure supplement 2.** Replicate ACT1-CUP1 assays to those shown in *Figure 7*.

accumulation was observed in the absence of Fyv6. This result suggests that Fyv6 could act as a negative regulator of splicing of the *SUS1* transcript when a Prp22 helicase domain mutant is present. Removing Fyv6 restores splicing by activation of the *SUS1* BP proximal 3′ SS.

To further explore the connections between Fyv6 and Prp22 fidelity, we used the ACT1-CUP1 assay in which yeast growth in the presence of increasing concentrations of $Cu^{2+}$ is proportional to the extent of splicing of a reporter RNA (***Figure 7F and G***; ***Figure 7—figure supplements 1 and 2***). Since the Prp22 R805A mutant grew poorly, we only assayed the Prp22 G810A, I1133R, and G810A/I1133R (double) mutants in the presence and absence of Fyv6. Consistent with previous results, we found that *fyv6Δ* is deleterious for splicing of the WT ACT1-CUP1 reporter containing consensus SS. Copper tolerance was improved when *fyv6Δ* was combined with Prp22 I1133R. Unexpectedly, Prp22 I1133R was deleterious when Fyv6 was present for this reporter. If this Prp22 mutant and Fyv6 both promote the second step, then their combination may have an additive, negative effect on splicing due to over-stabilization of this spliceosome conformation.

Finally, we assayed strains containing a 3′ SS UAG to gAG reporter since it was previously shown that Prp22 helicase domain mutants could increase usage of non-consensus 3′ SS due to faulty proof-reading activity (***Figure 7F and G***). In agreement with this, we observed a dramatic increase in copper tolerance when Prp22 contained both the C-terminal domain and helicase mutants (double). This suggests that splicing of non-consensus 3′ SS is enhanced both by stabilization of the second step conformation (I1133R) and perturbing mRNA release (G810A). This effect on Prp22 is Fyv6-dependent since loss of Fyv6 results in very poor copper tolerance of strains containing the gAG reporter, likely since this 3′ SS is located far from the BP (38 nt). These results show that *fyv6Δ* can be suppressed with a Prp22 C-terminal domain mutant and that mutations in the Prp22 C-terminal domain can have distinct functional consequences from those located in the helicase domain. In the case of *SUS1*, stalling of splicing due to the Prp22 G810A mutant can be relieved by removing Fyv6 and activation of a BP proximal 3′ SS.

## Discussion

Here, we establish Fyv6 as a second step splicing factor in yeast that promotes usage of BP distal 3′ SS. Structural, genetic, and biochemical studies reveal critical contacts and interactions between Fyv6 and the splicing machinery. From these data, we propose that Yju2/Syf1 interactions need to be disrupted for proper Fyv6 recruitment and that Fyv6 can enforce the Prp22 dependence of at least some splicing events. Combined, our data allow us to propose a model for Fyv6's role during the second step (***Figure 8***). In this model, Fyv6 is recruited to the C* complex after Prp16-dependent release of Yju2, which exposes the binding site for Fyv6 on Syf1. Once bound, Fyv6 (possibly in concert with Slu7 and Prp18) facilitates docking of BP distal 3′ SS into the active site. In the absence of Fyv6, the use of BP distal 3′ SS becomes more difficult and BP proximal 3′ SS are favored. Under these conditions, the requirement for Prp22 during exon ligation is reduced and Yju2 may also potentially re-bind Syf1. One or both of these factors could contribute to formation of a lower fidelity spliceosome with relaxed sequence constraints for exon ligation.

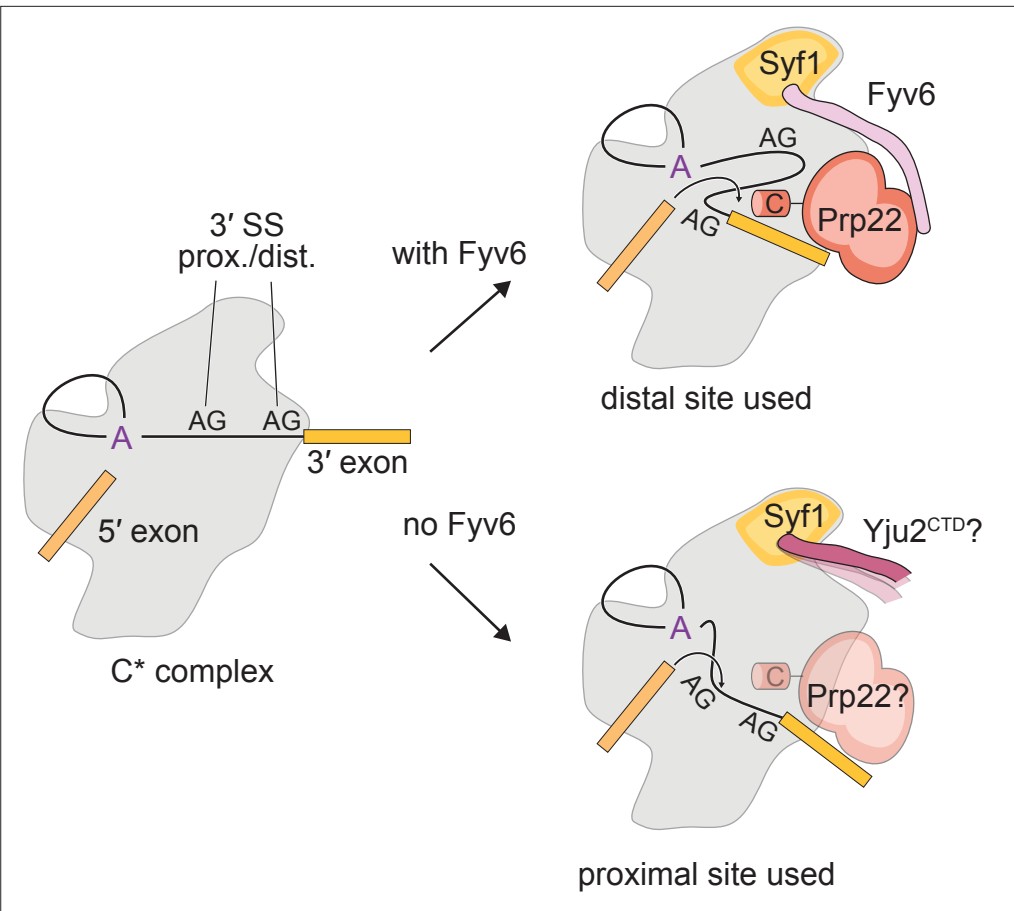

**Figure 8.** Model for the role of Fyv6 during the second step. Fyv6 interacts with Syf1 and Prp22 and promotes usage of BP distal 3' splice site (SS). When Fyv6 is absent, branch point (BP) proximal 3' SS are favored. It is possible that lack of Fyv6 also results in the presence of Yju2 during exon ligation and relaxing of the requirement for Prp22 at this step.

## Structural insights into the recruitment and release of second step factors

A key principle of spliceosome structural biochemistry is the presence of mutually exclusive interactions that coordinate the stepwise and directional progression of splicing (*Townsend et al., 2020*). Our cryo-EM structure of the P complex spliceosome suggests another such interaction is involved in the first to second step transition during which Fyv6 displaces Yju2. Yju2 and Fyv6 occupy overlapping binding sites on the splicing factor Syf1 and, in fact, unassigned density in spliceosome P complexes due to Fyv6 was previously mis-interpreted as Yju2 due to their structural similarity (*Wilkinson et al., 2017*; *Wilkinson et al., 2021*). We propose that Prp16-promoted spliceosome remodeling results in release of Yju2 from the spliceosome C complex and permits mutually exclusive binding of Fyv6. Fyv6 remains bound in the P complex but then may dissociate to permit re-binding of Yju2, which was observed by cryo-EM to be a component of the yeast ILS complex (*Wan et al., 2017*). It should be noted, however, that intron lariats only weakly co-immunoprecipitate with Yju2 (*Liu et al., 2007a*) and that Yju2's presence in the cryo-EM data likely originated from the ILS complexes being purified with a TAP-tag on Yju2 itself (*Wan et al., 2017*). Since Yju2 has also not been observed in metazoan ILS complexes (*Vorländer et al., 2024*), its role in spliceosome disassembly and any need for Fyv6-Yju2 exchange during the P to ILS transition should be studied further.

Previously, Chiang and Cheng noted that while the N-terminal half of Yju2 was able to restore first step activity to Yju2-depleted extracts, it is much more stably associated if a second polypeptide containing the C-terminal, Syf1-interacting half was included *Chiang and Cheng, 2013*. It is possible that competition between Yju2 and Fyv6 for Syf1 binding can explain these results. If only

the N-terminal half of Yju2 is present, Fyv6 could associate prematurely with Syf1 and result in destabilization of the first step conformation in favor of second step interactions. This may then result in weaker binding for the N-terminal half of Yju2. Failure to properly recruit and exchange the Yju2/Fyv6 interaction on Syf1 may also be the origin of some of the yeast growth and splicing phenotypes we observe (*Figure 5*).

The need for Yju2/Fyv6 exchange hints at two distinct functions for Fyv6: a role in Yju2 displacement to prevent its re-binding during exon ligation and a role in stabilization of the second step spliceosome conformation. These two functions can potentially result in the distinct *cs* and *ts* phenotypes in yeast. While more work is needed to dissect these phenotypes, it is possible that the *cs* phenotype results from increased stability of the Yju2/Syf1 interaction at lower temperatures and the need for Fyv6 to act as a competitor of this interaction. The *ts* phenotype on the other hand may result from destabilization of the second step conformation of the spliceosome active site due to loss of Fyv6. It is possible that loss of Fyv6 results in destabilization due to removal of a buttress that holds the RNase H-like domain of Prp8 in place (along with Prp18), changes in how the C-terminal domain of Prp22 interacts with the spliceosome active site, removal of an impediment to reversal of the U2 snRNA/BS duplex conformational change that occurs during the C to C* complex transition, or a combination of these factors.

The high quality of our cryo-EM data combined with 3D classification also provide new insights into the requirements for 3′ SS docking into the active site and potentially for how the P complex spliceosome is disassembled into the ILS complex. It was previously speculated that the N-terminal domain of Slu7 may be critical for 3′ SS docking by reducing the entropic cost of this process when the BP to 3′ SS distance is long (*Wilkinson et al., 2017*). In our state II P complex structure, the 3′ SS remains docked but this domain of Slu7 is not observed. This suggests that either Slu7 N-terminal domain interactions with Cwc22 and Prp8 are not essential for 3′ SS docking into the active site for exon ligation and/or that these interactions are not essential for maintaining 3′ SS docking within P complex once they form in C*.

In the state III P complex structure, densities for the second step factors Fyv6, Prp18, and Slu7 are entirely missing and the 3′ SS is no longer docked into the active site. While it is possible that this may reflect decomposition of the P complex (or another intermediate) by a pathway unrelated to cellular spliceosome disassembly, the structure is suggestive that 3′ SS undocking is coupled to second step factor release. These steps may be necessary to help prepare the spliceosome for ILS complex formation since the 3′ SS is undocked in all ILS structures obtained to date (*Vorländer et al., 2024*; *Wan et al., 2017*; *Yan et al., 2015*; *Zhang et al., 2019*). Prp22 is still present in this structure, and it is possible that state III represents a conformation occurring after exon ligation but prior to Prp22-dependent mRNA release. In this case, the state III structure implies that Prp22 ATPase activity is not needed for dissociation of Fyv6, Prp18, and Slu7. It is also interesting to note that the C-terminal domain of Prp22 can no longer be resolved in the state III structure. What exactly activates the Prp22 ATPase to promote proofreading or mRNA release is not known. It is conceivable that undocking of the 3′ SS from the active site can be sensed by the Prp22 C-terminal domain and this contributes to ATPase activation. This model differs from that previously proposed for Prp22-enabled 3′ SS sampling in which that ATPase activity of Prp22 is needed to undock the 3′ SS (*Semlow et al., 2016*). However, sampling could involve multiple steps: ATP-independent 3′ SS undocking and ATP-dependent entry into the sampling state. Whether or not the second step factors must release or alter conformation to enable sampling is unknown. Regardless of the exact mechanism, it is important to recognize that the P complex structures determined here were all obtained using a Prp22 mutant defective in mRNA release. The chemical and kinetic competencies of these states will require future investigations.

## Multiple, non-redundant second step factors promote use of BP distal 3′ SS

The challenge of 3′ SS selection involves not just selection of the correct nucleotide sequence (YAG) but also identifying which YAG sequence to use. Here, we define a function for Fyv6 in facilitating use of BP distal 3′ SS, especially those located ≥21 nt away from the branch. This function is similar to those of other second step factors important for efficient splicing when the BP-3′ SS distance is ≥12 nt (Slu7, Prp18) or ≥21 nt (Prp22) (*Brys and Schwer, 1996*; *Schwer and Gross, 1998*; *Zhang and Schwer, 1997*). Human Slu7 has additionally been reported to be important for preventing suppression of the

correct 3′ SS in favor of up- or downstream alternative 3′ SS (*Chua and Reed, 1999*). Our previous study using competing 3′ SS in the ACT1-CUP1 intron did not provide evidence for suppression of the correct 3′ SS in the absence of Fyv6, suggesting that this function is not shared between human Slu7 and yeast Fyv6. Nonetheless, it is important to recognize that the splicing machinery employs multiple factors that enforce usage of BP distal 3′ SS. These functions are not redundant as changes in 3′ SS are still observed in the absence of Fyv6 but presence of Prp18 and vice versa (*Roy et al., 2023*).

Intron architecture can also contribute to 3′ SS selection since canonical 'AG' 3′ SS are depleted in humans in the region between the BP and annotated 3′ SS (the AG exclusion zone [AGEZ]) (*Corvelo et al., 2010*; *Gooding et al., 2006*). AG sites are also typically depleted in the corresponding region in yeast introns (data not shown). This would suggest that annotated 3′ SS are also the most optimal sites located downstream of the BP. However, an important assumption of this model is that non-AG 3′ SS can also be effectively proofread and not used. Deletion of either Prp18 or Fyv6 leads to activation of not just BP proximal 3′ SS but also those with highly divergent sequences from the YAG consensus (*Roy et al., 2023*). This suggests that an AGEZ alone is insufficient for correct 3′ SS usage and that mechanisms to ensure preferential use of YAG sites are critical.

It is likely that Fyv6 and Prp18 enforce usage of YAG sites by common as well as factor-specific mechanisms. Both proteins stabilize the conformation of the Prp8 RNase H-like domain in P complex. Numerous SS suppressor mutations map to this domain including those that enhance splicing of non-YAG 3′ SS (*Galej et al., 2013*). Fyv6 and Prp18 may be essential for establishing a 'high fidelity' conformation of the RNase H-like domain that enforces use of canonical YAG 3′ SS and preventing activation of cryptic sites within the AGEZ. In support of this, we note that in our state III P complex structure that lacks both Fyv6 and Prp18 and a stably docked 3′ SS, the RNase H-like domain is also more flexible.

In addition to this possibility, Fyv6 and Prp18 may also contribute uniquely to 3′ SS fidelity and selection. The conserved loop region of Prp18 inserts into the spliceosome active site and contacts the 3′ SS. Partial deletion of this region of Prp18 phenocopies at least some of the changes in 3′ SS selection when Prp18 is deleted entirely (*Crotti et al., 2007*; *Roy et al., 2023*). This indicates that even when the Prp8 RNaseH-like domain is still stabilized by Prp18 (and Fyv6) disruptions to 3′ SS selection can still occur. Unlike Prp18, Fyv6 does not contact the spliceosome active site. However, it is the only second step factor that contacts exon ligation fidelity factor Prp22. While these contacts are not essential for suppressing use of an alternative 3′ in *SUS1*, a mutation in Prp22 can nonetheless suppress a *fyv6Δ* phenotype and loss of Fyv6 relieves a Prp22 helicase mutant-dependent stall in *SUS1* splicing. These observations suggest that Fyv6 and Prp22 are functionally connected to one another.

## Some second step factors may enforce usage of Prp22 during exon ligation

We were particularly struck by the similarities in the BP-3′ SS distances for the requirement of Prp22 during exon ligation and for promotion of BP distal 3′ SS usage by Fyv6, both ≥21 nt (*Schwer and Gross, 1998*). In addition, the alternative, BP proximal 3′ SS that are activated when Fyv6 is deleted show much less sequence conservation relative to the annotated 3′ SS and include sequences (GAG, BG) known to be proofread and rejected by Prp22. Together, this suggests that one function of Fyv6 is to enforce Prp22 dependence of exon ligation by stabilizing a spliceosome conformation that more easily enables usage of BP distal 3′ SS (*Figure 8*). As these sites require Prp22 for exon ligation, it is also likely that they are subject to Prp22-dependent proofreading, and this may enforce accurate exon ligation at the YAG 3′ SS consensus. When Fyv6 is absent, many alternative, BP proximal 3′ SS are activated. These sites could have a relaxed requirement for Prp22 and proofreading. While the Fyv6 dependence of Prp22 occupancy is beyond the scope of the current manuscript, loss of Prp18 (which also activates usage of non-canonical, BP proximal 3′ SS) does result in lower occupancy of Prp22 on the *ACT1* intron (*Strittmatter et al., 2021*). This suggests that Prp22 occupancy, and possible involvement in the exon ligation step, can be influenced by second step factors.

There is precedent for nonessential splicing factors enforcing use of a proofreading ATPase. Cus2 is not essential for pre-spliceosome formation (*Yan et al., 1998*). However, when Cus2 is present then ATPase activity of Prp5 is essential for disruption of the Hsh155/Cus2 interaction (*Perriman et al., 2003*; *Talkish et al., 2019*). In this case, Cus2 is not essential for Prp5 proofreading of BP selection, and the connection between the ATP hydrolysis activity of Prp5, removal of Cus2, and proofreading is unclear (*Liang and Cheng, 2015*; *Talkish et al., 2019*; *Xu and Query, 2007*). It is possible that

enforcement of Prp5 ATPase activity by Cus2 is important for other aspects of pre-spliceosome formation in vivo and the impact of Prp5 and Cus2 on BP selection has not yet been studied transcriptome-wide using high-resolution, sequencing-based approaches. The C-terminal domain of the first step splicing factor Yju2 (the same domain that competes with Fyv6 for Syf1 binding) is also not essential for 5′ SS cleavage. In the absence of this domain, the N-terminal domain of Yju2 is sufficient for promoting the first step reaction, and the spliceosome can transition to the second step and carry out exon ligation without Prp16, albeit inefficiently (*Chiang and Cheng, 2013*). Importantly, addition of the C-terminal domain on a separate polypeptide reduces Prp16-independent activity. This suggests that the Yju2 C-terminal domain enforces the use of the Prp16 ATPase. As we propose with Fyv6 and Prp22, the C-terminal domain of Yju2 may not just help to promote the first to second step transition by enforcing Prp16 dependence but could also facilitate Prp16-dependent proofreading during 5′ SS cleavage (*Koodathingal et al., 2010*; *Semlow et al., 2016*). General operating principles of the yeast splicing machinery may not just include proofreading but also use of splicing factors that enforce usage of the proofreading ATPases.

# Materials and methods

## Key resources table

| Reagent type (species) or resource | Designation | Source or reference | Identifiers | Additional information |
|---|---|---|---|---|
| Strain, strain background (*Saccharomyces cerevisiae*) | Various *S. cerevisiae* strains | Various sources, see *Supplementary file 6* for a full list | | See *Supplementary file 6* for full list |
| Recombinant DNA reagent | Various plasmids | Various sources, see *Supplementary file 7* for a full list | | See *Supplementary file 7* for full list |
| Sequence-based reagent | Various oligonucleotides | Integrated DNA Technologies Inc (IDT); see *Supplementary file 8* for additional details | | See *Supplementary file 8* for full list |
| Antibody | Anti-FLAG (Mouse monoclonal) | Sigma-Aldrich | Cat #: F1804; RRID:AB_262044 | WB (1:1000) |
| Antibody | Anti-actin (mouse monoclonal) | Sigma-Aldrich | Cat #: MAB1501; RRID:AB_2223041 | WB (1:5000) |
| Antibody | Anti-mouse IgG – HRP (Goat polyclonal) | Bio-Rad | Cat #: 1706516; RRID:AB_11125547 | WB (1:8000) |
| Peptide, recombinant protein | Bovine Serum Albumin | Sigma | Cat#: A7906 | |
| Commercial assay or kit | Clarity Western ECL substrate | Bio-Rad | Cat #: 1705060 | |
| Commercial assay or kit | Monarch RNA Cleanup Kit (50 µg) | New England Biolabs | Cat #: T2040 | |
| Commercial assay or kit | TURBO DNA-free kit | Thermo Fisher Scientific | Cat #: AM1907 | |
| Commercial assay or kit | TruSeq Stranded mRNA kit | Illumina | Cat #: 20020594 | |
| Commercial assay or kit | MasterPure Yeast RNA Purification kit | LGC Biosearch Technologies | Cat #: MPY03100 | |
| Commercial assay or kit | Access RT-PCR System | Promega | Cat #: A1250 | |
| Chemical compound, drug | Ni-NTA agarose | QIAGEN | Cat #: 30210 | |
| Chemical compound, drug | Amylose resin | New England Biolabs | Cat #: E8022S | |
| Chemical compound, drug | Fluorescein-5-thiosemicarbazide | Sigma-Aldrich | Cat #: 46985-100MG-F | |
| Software, algorithm | RELION-5.0 | *Kimanius et al., 2024* | RRID:SCR_016274 | https://relion.readthedocs.io/en/latest/index.html |
| Software, algorithm | Topaz | *Bepler et al., 2019* | | https://cb.csail.mit.edu/topaz/ |
| Software, algorithm | Coot | *Casañal et al., 2020* | RRID:SCR_014222 | https://www2.mrc-lmb.cam.ac.uk/personal/pemsley/coot/ |
| Software, algorithm | ISOLDE | *Croll, 2018* | RRID:SCR_025577 | https://tristanic.github.io/isolde/ |
| Software, algorithm | PHENIX | *Liebschner et al., 2019* | RRID:SCR_014224 | https://phenix-online.org/ |
| Software, algorithm | UCSF ChimeraX | *Pettersen et al., 2021* | RRID:SCR_015872 | http://www.cgl.ucsf.edu/chimerax/ |

*Continued on next page*

*Continued*

| Reagent type (species) or resource | Designation | Source or reference | Identifiers | Additional information |
|---|---|---|---|---|
| Software, algorithm | R Project for Statistical Computing | R Foundation for Statistical Computing | RRID:SCR_001905 | https://www.r-project.org/ |
| Software, algorithm | FASTQC | *Andrews, 2010* | RRID:SCR_014583 | http://www.bioinformatics.babraham.ac.uk/projects/fastqc/ |
| Software, algorithm | Fastp | *Chen et al., 2018* | RRID:SCR_016962 | https://github.com/OpenGene/fastp |
| Software, algorithm | STAR | *Dobin et al., 2013* | RRID:SCR_004463 | https://github.com/alexdobin/STAR/releases |
| Software, algorithm | SAMtools | *Li et al., 2009* | RRID:SCR_002105 | https://www.htslib.org/ |
| Software, algorithm | Salmon | *Patro et al., 2017* | RRID:SCR_017036 | https://combine-lab.github.io/salmon/ |
| Software, algorithm | DeSeq2 | *Love et al., 2014* | RRID:SCR_015687 | https://bioconductor.org/packages/release/bioc/html/DESeq2.html |
| Software, algorithm | featureCounts | *Liao et al., 2014* | RRID:SCR_012919 | https://bioconductor.org/packages/release/bioc/html/Rsubread.html |
| Software, algorithm | SpliceWiz | *Wong et al., 2023* | | https://github.com/alexchwong/SpliceWiz |
| Software, algorithm | Image J 1.53v | *Schneider et al., 2012* | RRID:SCR_003070 | https://imagej.net/ij/ |
| Other | Cryo-EM grid | Quantifoil | Cat #: Q350CR2-2NM | Used to prepare samples for cryo-EM |
| Other | NuPAGE Bis-Tris mini protein gel | Invitrogen | Cat #: NP0322 | Used for SDS-PAGE analysis of proteins and western blotting |
| Other | NuPAGE MES SDS Running Buffer | Invitrogen | Cat #: NP0002 | Used for SDS-PAGE analysis of proteins and western blotting |
| Other | Standard Vitrobot Filter Paper, Ø55/20 mm, Grade 595 | Ted Pella | Cat #: 47000-100 | Used to prepare samples for cryo-EM |
| Other | Mixed cellulose ester filter paper | Whatman | WHA7140104 | Used for yeast isolation |
| Other | TRIzol | Thermo Fisher Scientific | Cat #: 15596018 | Used for RNA extraction from yeast |
| Other | SuperScript III reverse transcriptase | Thermo Fisher Scientific | Cat #: 18080044 | Used for RT-PCR |
| Other | Zymolyase | Zymo Research | Cat #: E1005 | Used at 50 U/mL; used for RNA extraction from yeast |

## Yeast growth and culture

Yeast transformation, plasmid shuffling/5-FOA selection, and growth were carried out using standard techniques and media (*Treco and Lundblad, 1993*; *Sikorski and Boeke, 1991*).

## Cryo-electron microscopy

### Purification of dominant-negative Prp22

Yeast Prp22 with a dominant-negative S635A mutation was cloned into pRS424 and pRS426 vectors after an N-terminal CBP-His-TEV tag (*Galej et al., 2013*). The protein was expressed in 24 L of BCY123 yeast cells essentially as described (*Galej et al., 2013*). Recombinant Prp22 was then purified first by Calmodulin-Sepharose in a buffer containing 50 mM Tris-HCl pH 8.5, 350 mM NaCl, 2 mM CaCl$_2$, 1 mM Mg acetate, 1 mM imidazole, 0.1% NP-40 substitute, 5 mM β-mercaptoethanol, and eluted with 3 mM EGTA. The protein was dialyzed into 20 mM Tris-HCl pH 7.4, 350 mM NaCl, 10 mM imidazole, 0.1% NP-40 substitute and further purified over Ni-NTA resin at 500 mM NaCl and eluted with 200 mM imidazole. The eluted protein was treated with His-tagged TEV protease during dialysis against 20 mM HEPES-KOH pH 7.9, 300 mM KCl, 0.2 mM EDTA, 20% glycerol, 0.5 mM DTT, before removal of the protease with Ni-NTA resin. The final protein was frozen in liquid nitrogen and stored at –80°C.

## Preparation and purification of P complex for cryo-EM

P complex was purified as described in *Wilkinson et al., 2017*. Brr2-TAPS yeast were grown in a 120 L fermenter and splicing extract was prepared by the liquid nitrogen method essentially as previously described (*Lin et al., 1985*). A pre-mRNA substrate consisting of 20 nt 5'-exon, 95 nt intron, and 32 nt 3'-exon from the UBC4 pre-mRNA followed by 3xMS2 stem loops was generated by run-off in vitro transcription from an EcoRI-linearized plasmid template. The RNA product was labeled at the 3' end with fluorescein-5-thiosemicarbazide (*Wu et al., 1996*). A 180 mL in vitro splicing reaction was performed containing 60 mM potassium phosphate pH 7, 2 mM ATP, 2.5 mM MgCl$_2$, 3% wt/vol PEG8000, 3 nM pre-mRNA, 37.5 nM MS2-MBP protein, and 40% (vol/vol) splicing extract treated with dominant negative Prp22 S635A. Reactions were incubated for 30 min at 23°C, then incubated for 20 min with 5 mM of a DNA oligonucleotide complementary to the 3'-exon (oligo sequence 5'-ATGA AGTAGGTGGAT-3') to induce cleavage of the 3'-MS2 tag by the endogenous RNase H activity of the splicing extract. The reaction mixture was centrifuged through a 40% glycerol cushion in buffer A (20 mM HEPES, pH 7.9, 75 mM KCl, 0.25 mM EDTA). The cushion was collected and applied to amylose resin in the presence of 0.025% NP-40 substitute. After overnight incubation at 4°C the resin was washed and eluted with buffer A containing 5% glycerol, 0.01% NP-40 substitute, and 12 mM maltose. Fractions containing spliceosomes were concentrated to 0.1 mg/mL then dialyzed against buffer A for 3 hr.

For cryo-EM grid preparation, a freshly glow-discharged Cu300 R2/2 holey carbon grid with a 2 nm layer of amorphous carbon (Quantifoil) was mounted in the chamber of a Vitrobot Mark IV (Thermo Fisher Scientific) maintained at 4°C and 100% humidity. Spliceosomes (3 µL) were applied and after 60 s was manually blotted using Ø55 grade 595 filter paper (Ted Pella), followed by application of another 3 µL of spliceosomes, waiting 60 s, before blotting again and plunging into liquid ethane.

## Cryo-EM data collection

Cryo-EM data were collected as two datasets from two grids made from the same sample. Dataset 1 was acquired using a Thermo Scientific Titan Krios cryo-TEM (LMB Krios 1) and dataset 2 was acquired using a Thermo Scientific Titan Krios G3i cryo-TEM (LMB Krios 3). Both used a K3 direct detector (Gatan) operated in super-resolution mode with twofold binning, and an energy filter with slit width of 20 eV. Micrographs were collected automatically using EPU in AFIS mode, yielding a total of 51,113 movies at ×81,000 or ×130,000 magnification with a real nominal pixel size of 0.93 Å or 0.669 Å, with defocus ranging from –1.3 µm to –3.1 µm and a total fluence per micrograph of 40 e⁻/Å², fractionated into 40 frames.

## Cryo-EM data processing

All cryo-EM data were processed using RELION-5.0 (*Kimanius et al., 2024*). Both datasets were initially processed separately. Movies were corrected for motion using the RELION implementation of MotionCor2, with 4×4 patches and dose-weighting. CTF parameters were estimated using CTFFIND-4.1. Particle picking was done using Topaz with the general model (*Bepler et al., 2019*), yielding 1,820,457 particles. 3D classification was performed on fourfold (dataset 1) or fivefold (dataset 2) binned particles using a P complex reference map (EMD-10140), and 299,741 (dataset 1) or 599,123 (dataset 2) were selected and refined to 2.99 Å or 2.78 Å resolution respectively. After Bayesian polishing and CTF refinement (per-particle defocus, anisotropic magnification, beam tilt, trefoil, and fourth-order aberrations) the resolution improved to 2.67 Å or 2.32 Å, respectively. To improve the general density quality, 3D classification without alignment was performed with a soft mask around the whole complex with T=15. Particles with good density, without selecting for Fyv6 occupancy, were selected (151,443 and 257,501 for datasets 1 and 2) and refined to 2.66 Å or 2.27 Å, respectively. The particles were then merged, refined, and then refined for anisotropic magnification. The overall scaling factors from the 'anisotropic' magnification matrix was then used to calculate the corrected pixel size for dataset 1 as 0.94057 Å, keeping dataset 2 constant at 0.669 Å per pixel. Defocus values for dataset 1 were then scaled by the square of 0.94057/0.93, with an empirical constant correction of –10.06 Å (*Wilkinson et al., 2019*). Finally, another round of per-particle defocus refinement and anisotropic magnification refinement was performed to reduce errors in defocus correction and dataset scaling. The combined particles refined to 2.24 Å resolution. Resolution is reported using the gold-standard Fourier shell correlation with 0.143 cutoff.

The refined particles after merging were subjected to 3D classification without alignment, with a soft mask around the entire spliceosome, with T=15. The resulting four classes were used to define States I, II, and III. To improve the densities for the periphery of the spliceosome that displayed local flexibility, focused refinements were used with Blush regularization in RELION-5 to reduce overfitting (*Kimanius et al., 2024*). The Prp22 density was improved using a soft mask just around the Prp22 helicase domain for refinement of particles from States I and II, yielding a 2.96 Å map. Densities for the U2 snRNP, NTC, U5 Sm ring, and Cwc22 N-terminal domain were each improved first by 3D classification without alignment of particles from all three states, selecting particles with strong density (72,869 for the U2 snRNP, 260,409 for the NTC, 113,561 for the U5 Sm ring, 201,984 for Cwc22-NTD), performing signal subtraction to remove density for the rest of the spliceosome (recentering on the mask center of mass and re-boxing to 300 pixels, or 192 pixels for Cwc22-NTD), and finally 3D refinement starting with 1.8 degree local angular searches, giving 3.42 Å for the U2 snRNP, 3.72 Å for the NTC, 3.06 Å for the U5 Sm, and 3.57 Å for Cwc22-NTD. In all cases, control refinements without Blush regularization did not produce interpretable maps (*Figure 3—figure supplement 4*).

## Model building

The model was built using Coot (*Casañal et al., 2020*) and ISOLDE (*Croll, 2018*) starting with PDB 6EXN (*Wilkinson et al., 2017*) as an initial model for the core and PDB 7B9V (*Wilkinson et al., 2021*) as an initial model for the periphery. The core was improved by manual fixing in Coot, while the peripheral U2 snRNP and NTC were improved by applying torsion and distance restraints to reference models produced by AlphaFold2. The model was refined using PHENIX real_space_refine (*Liebschner et al., 2019*), just performing one macro-cycle. Figures were generated using UCSF ChimeraX.

## Plasmid cloning

Fyv6 plasmids were made by PCR amplification from gDNA of the *FYV6* coding sequence plus ~250 nt upstream and downstream, restriction digest with NotI and SalI, and ligation into pRS414 or pRS416 (*Mumberg et al., 1995*). An N-terminal 1x FLAG tag was added onto Fyv6 in the plasmids by PCR mutagenesis. Fyv6 truncations were cloned by Genewiz (Azenta Life Sciences) from pAAH1572. Restriction digest and ligation were used to place Fyv6 constructs into the pRS413 backbone.

Syf1 plasmids were made by PCR amplification from gDNA of the SYF1 coding sequence plus ~275 nt upstream and downstream, restriction digest, and ligation into pRS414 or pRS416.

Cef1 mutant plasmids, Prp22 I1133R mutant plasmid, Prp8 mutant plasmids, ACT1-CUP1 BP-3′ SS distance reporters, and Syf1 truncation plasmids were prepared by PCR-based site-directed mutagenesis of pAAH1611, pAAH1042, pAAH1440, pAAH0470, and pAAH1625, respectively. All plasmids were fully sequenced to confirm their nucleotide sequence.

## Yeast strain creation

The *DBR1* and *FYV6* genes were deleted by replacement with a nourseothricin resistance cassette (NatMX4) or hygromycin resistance cassette (hphMX4), respectively, through homologous recombination (see *Supplementary file 7*; *Goldstein and McCusker, 1999*). The *SYF1* and *CEF1* genes were deleted by replacement with a kanamycin resistance cassette (KanMX) through homologous recombination in strains transformed with pRS416-Syf1 or pRS316-Cef1 respectively. Gene deletion was confirmed by colony PCR and/or genomic DNA extraction from the strains and PCR amplification of the corresponding genomic locus.

## RNA isolation

Yeast were grown in YPD media until $OD_{600}$ 0.5–0.8. For RNA-seq of temperature-shifted samples, cultures were shifted to either 37°C or 16°C or remained at 30°C and grown for 1 hr. A 15 mL volume of culture was harvested via vacuum filtration with mixed cellulose ester filter paper (Whatman, 47 mm diameter) in a Buchner funnel. Filter papers were separated into 5 mL tubes and flash-frozen. Filters were stored at –80°C until RNA isolation. Filters were washed with 1 mL TRIzol to collect cells. For all other experiments involving RNA isolation, 10 $OD_{600}$ units of $OD_{600}$ 0.5–0.8 yeast were harvested via centrifugation.

Cell walls were disrupted by incubation with Zymolyase (50 U/mL, Zymo Research) in Y1 buffer (1 M sorbitol, 100 mM EDTA pH 8.0, 13 mM β-mercaptoethanol) (primer extension, RNA-seq of

non-temperature-shifted samples) or vortexing with silica disruption beads in TRIzol (RT-PCR, RNA-seq of temperature-shifted samples). Total RNA was then isolated using TRIzol reagent (Thermo Fisher Scientific) according to the manufacturer's instructions. Samples were cleaned using a 50 µg Monarch RNA Cleanup Kit (New England Biolabs) and treated with TURBO DNase (Thermo Fisher Scientific) to remove residual contaminating DNA. For RNA-seq, quality of RNA samples was assessed prior to library preparation by NanoDrop (concentration, $A_{260}/A_{280}$, and $A_{260}/A_{230}$), Qubit (RNA High Sensitivity and RNA IQ), and TapeStation (RNA ScreenTape Analysis). High-quality samples showing minimal rRNA degradation with RIN>7 and A260/A280 ratios of ~2.0 were selected for library preparation.

## Library preparation and RNA-seq

Library preparation was conducted by the University of Wisconsin-Madison Biotechnology Center Gene Expression Center. mRNA was selected from total RNA samples through polyA enrichment with the TruSeq Stranded mRNA kit (Illumina). Sequencing was performed with an NovaSeq6000 sequencer (Illumina).

## Bioinformatic analysis of RNA-seq datasets

FASTQC was used for quality control of reads pre- and post-trimming (*Andrews, 2010*). Trimming of reads was accomplished using fastp (*Chen et al., 2018*) prior to mapping to the SacCer3 genome (Ensembl, R64-1-1) with STAR (*Dobin et al., 2013*). All samples were aligned with STAR in a first pass. Novel junctions from all samples (WT and *fyv6Δ*) were combined and filtered for likely false-positive junctions (see Materials and methods, 'Read mapping'). STAR was run in a second pass with the additional input of the filtered set of novel junctions. Indexing of bam files was accomplished with SAMtools (*Li et al., 2009*). Quantitation of read counts aligned to the SacCer3 transcriptome was accomplished with Salmon (*Patro et al., 2017*). Differential gene expression analysis was conducted with DeSeq2 (*Love et al., 2014*). Junction reads within genes were counted with featureCounts (*Liao et al., 2014*) and used to calculate FAnS based on methods in *Roy et al., 2023* (see Supplemental Methods). Canonical BP were used as defined in the Ares Intron Database (*Grate and Ares, 2002*). Differential splicing analysis was conducted with SpliceWiz (*Wong et al., 2023*).

The following flags were used with the specified tools for RNA-seq data analysis.

### Assessment of RNA-seq quality
FASTQC.

### Trimming
fastp `--detect_adapter_for_pe` -Q -q 20 -u 40 -l 36 `--poly_g_min_len` 10 -g -5 -3 -W 4 -M 20.

### Saccer3 annotation
Ensembl, R64-1-1.

### Read mapping
All samples were aligned with STAR in a first pass. Novel junctions from all samples (WT and *fyv6Δ*) were combined and filtered for likely false-positive junctions (non-canonical junctions, column 5 >0; junctions supported by multi-mappers only, column 7>0; junctions supported by two few reads, column 7>2). STAR was run in a second pass with the additional input of the filtered set of novel junctions as follows: `--runMode alignReads --runThreadN 20 --genomeDir./genome --sjdb-GTFfile` annotation.gtf `--alignIntronMin` 10 `--alignIntronMax` 2000 `--readFilesCommand zcat --readFilesIn` 'R1_trimmed.fastq.gz' 'R2_trimmed.fastq.gz' `--outSAMtype` BAM SortedByCoordinate `--limitBAMsortRAM` 6000000000.

### SpliceWiz differential splicing analysis
with the following flags: rmats.py -t paired `--readLength` 150 `--variable-read-length` `--novelSS`.

## FeatureCounts split read counting

featureCounts -a annotation.gtf -o Sample `--splitOnly` -J -f -p -T 5 -R BAM -B -C Aligned.sorted-ByCoord.out.bam.

## Salmon read count quantitation

Salmon was run on the free resource usegalaxy.org with the following flags: `--libType` A `--incompatPrior` '0.0' `--biasSpeedSamp` '5' `--fldMax` '1000' `--fldMean` '250' `--fldSD` '25' `--forgettingFactor` '0.65' `--maxReadOcc` '100' `--numBiasSamples` '2000000' `--numAuxModelSamples` '5000000' `--numPreAuxModelSamples` '5000' `--numGibbsSamples` '0' `--numBootstraps` '0' `--thinningFactor` '16' `--sigDigits` '3' `--vbPrior` '1e-05'. Differential gene expression analysis was conducted with DeSeq2 with the following flags: -t 1 -P -V 10 -i -y salmon -x mapping.gff.

## DeSeq2 differential gene expression analysis

deseq2.R -o 'output.dat' -p -A 0.1 -H -f '[["Fyv6", [{"WT": ["WT1.tabular", "WT2.tabular"]}, {"Fyv6": ["Fyv61.tabular", "Fyv62.tabular"]}]]]' -l '{"Fyv61.tabular": "Fyv61.tabular", "Fyv62.tabular": "Fyv62.tabular", "WT1.tabular": "WT1.tabular", "WT2.tabular": "WT2.tabular"}' -t 1 -P -V 10 -i -y salmon -x.

## FAnS calculation

Junction reads within genes were counted with featureCounts. Genes were filtered based on those listed in the Ares Intron database. Canonical 5′ and 3′ SS were annotated based on the highest number of junction counts. All junctions were filtered based on presence in all four RNA-seq samples. Read counts for filtered junctions were combined for *fyv6Δ* and WT replicates. Alternative junctions were separated into those that shared a canonical 5′ SS and an alternative 3′ SS or those that shared a canonical 3′ SS and an alternative 5′ SS. FAnS was calculated based on the number of junction reads for a unique alternative 3′ SS sharing a canonical 5′ SS divided by the number of junction reads for canonical 5′ SS and 3′ SS within a sample to adjust for changes in expression. Ratios of FAnS were calculated by dividing the FAnS value for *fyv6Δ* by the FAnS value for WT.

## Docker images

    fastp – biocontainers/fastp
    STAR – alexdobin/star:2.7.10a_alpha_220506
    featureCounts – pegi3s/feature-counts:2.0.0
    Salmon – combinelab/salmon:1.10.3
    FASTQC – jysgro/fastqc:ub2306_12.1

## Primer extension

RNA was isolated using the protocol above. IR700 dye conjugated probes (Integrated DNA Technologies) were used for primer extension with SuperScript III (Thermo Fisher) of the ACT1-CUP1 reporters (2 pmol yAC6: /5IRD700/GGCACTCATGACCTTC) and U6 snRNA (0.4 pmol yU6: /5IRD700/GAACTGCTGATCATGTCTG) (*Carrocci et al., 2017*; *van der Feltz et al., 2021*). Primer extension products were visualized on a 7% (wt/vol) denaturing polyacrylamide gel run at 35 W for 80 min at room temperature. Gels were imaged with an Amersham Typhoon NIR laser scanner (Cytiva), and band intensities were quantified with ImageJ (version 1.53v, 2022).

## RT-PCR

Yeast cultures were inoculated from cultures grown to stationary phase overnight and grown until $OD_{600}$=0.7–0.9. RNA was isolated and depleted of contaminating DNA using the MasterPure Yeast RNA Purification Kit (LGC Biosearch Technologies) protocol as previously described (*Carrocci et al., 2017*) (RT-PCR shown in *Figure 4C*) or using the RNA isolation protocol above (all others).

RT-PCRs were set up using the Access RT-PCR system (Promega Corporation) following the kit procedure with 100 ng of input RNA per 25 μL reaction. Primers to amplify *SUS1* were SUS1-exon1 5′-TGGATACTGCGCAATTAAAGAGTC-3′ and SUS1-exon3 5′-TCATTGTGTATCTACAATCTCTTC

AAG-3′(*Hossain et al., 2009*). Reaction products were separated on 2% (wt/vol) agarose-TBE gels and imaged using ethidium bromide fluorescence.

## Selection of *fyv6Δ* suppressors

Strain yAAH3353 was previously described (*Lipinski et al., 2023*). Twenty single colonies of yAAH3353 were selected from a YPD plate and used to inoculate 20×5 mL YPD liquid culture. Cultures were grown for 48 hr at 30°C. A 1 mL volume of culture was spun down, resuspended in 200 µL YPD liquid medium, and spread on a YPD plate containing 40 mg/L adenine hemisulfate (to prevent ade2 reversion). Duplicate plates for each overnight were placed in incubators at either 18°C or 37°C. Many colonies of various sizes appeared on each plate by 9 days for 37°C and 12 days for 18°C. Colonies were picked after that time and annotated first by growth temperature, then by overnight number, and finally roughly by colony size with 01 generally being the largest colony on a plate (ex: 370101 from a plate grown at 37°C inoculated from overnight 01 and the largest colony selected from the plate). Picked colonies were grown overnight in 5 mL YPD at 30°C and frozen stocks were made. Anywhere from 0 to ~15 colonies were picked per plate, covering the variety of colony sizes that arose.

## Sequencing of yeast genomic DNA

Yeast were grown to saturated stationary phase in 5 mL YPD and ~10 OD units were sent as a frozen yeast pellet to Azenta or SeqCoast for genomic DNA extraction, library, preparation, and sequencing with 30×/400 Mbps coverage.

## Bioinformatic analysis of WGS data

Azenta and SeqCoast utilized the SacCer3 genome (Ensembl, R64-1-1) for variant calling and provided a list of >6000 SNPs. Since the strain is not a perfect match for this reference, non-unique SNPs were filtered from each dataset by taking the union of two sets with each set obtained from colonies isolated on different plates (to reduce the chance of computationally eliminating the same variant SNP). Filtering reduced the dataset to <100 unique variants per colony. Few unique SNPs were within protein- or snRNA-coding genes. Of SNPs within protein-coding regions, single amino acid changes were frequently found within splicing factors (see *Supplementary file 5*). Datasets were reexamined upon identification of suppressor mutations for suppressor mutation duplication that were potentially filtered out during comparison of SNP datasets.

## Yeast temperature growth assays

Yeast were grown to stationary phase overnight in the appropriate liquid medium indicated in the figure legends. The cultures were first diluted to $OD_{600}$ 0.5, serially diluted 1:10, 1:100, and 1:1000, and stamped onto agar plates containing the appropriate growth medium. The plates were incubated at 16°C, 23°C, 30°C, and 37°C, and plates were imaged after the number of days indicated in each figure.

## ACT1-CUP1 copper tolerance assays

Yeast strains expressing ACT1-CUP1 reporters were grown to stationary phase in –Leu DO media, diluted to $OD_{600}$=0.5 in 10% (vol/vol) glycerol, and spotted onto –Leu DO plates containing 0–2.5 mM $CuSO_4$ (*Lesser and Guthrie, 1993*; *Carrocci et al., 2018*). Plates were scored and imaged after 48 hr of growth at 30°C for WT strains and after 72 hr of growth at 30°C for *fyv6Δ* strains due to differential growth between strains.

## Western blotting

Total soluble protein was isolated by TCA precipitation from 10 ODs of an overnight yeast culture. Proteins were separated with a NuPAGE 4–12% Bis-Tris pre-cast gel (Invitrogen) using 1x NuPAGE MES running buffer (Invitrogen) and transferred to a PVDF membrane (0.2 µm pore size) at 30 V for 120 min at 4°C using 1x NuPAGE Transfer Buffer (Invitrogen) with 20% (vol/vol) methanol in a XCell II Blot Module (Invitrogen). The membrane was blocked with 3% (wt/vol) BSA (Sigma, A7906) in 1x TBST for 1 hr at room temperature with gentle shaking. All antibodies were diluted in 1x TBST with 3% (wt/vol) BSA. The primary antibodies anti-FLAG (MilliporeSigma, F1804; diluted 1:1000) and anti-actin (Sigma-Aldrich, MAB1501; diluted 1:5000) were incubated overnight at 4°C. The membrane was then

washed with 1x TBST before incubation with goat anti-mouse IgG HRP conjugate secondary antibody (Bio-Rad, #1706516; diluted 1:8000) for 2 hr at room temperature. The membrane was then washed with 1x TBST before developing with Clarity Western ECL substrate (Bio-Rad, #1705060) with a 20 s exposure for anti-actin and a 100 s exposure for anti-FLAG.

## Acknowledgements

We thank David Brow for his advice on the suppressor screen. We thank Justin Mabin for his advice on RNA-seq data analysis. MEW is grateful to Feng Zhang for funding and support, and to the labs of Kiyoshi Nagai and Kelly Nguyen for reagents and support. The authors utilized the University of Wisconsin – Madison Biotechnology Center Gene Expression Center (Research Resource Identifier – RRID:SCR_017757) for preparation of NGS libraries for RNA-seq and the Biochemistry Computational Research Facility (BCRF) for analysis of RNA-seq datasets. This work was supported by grants from the National Institutes of Health (R35 GM136261 to AAH), NIH Biotechnology Training Program (T32 GM135066), and National Science Foundation Graduate Research Fellowship Program (Grant No. DGE-1747503) fellowships to KAS, a Helen Hay Whitney Postdoctoral Fellowship (to MEW). This study was supported by the MRC Laboratory of Molecular Biology electron microscopy facility. We thank Guillaume Chanfreau for advice and Florian Heyd and Tucker Carrocci for critical reading of the manuscript.

## Additional information

### Funding

| Funder | Grant reference number | Author |
| --- | --- | --- |
| National Institutes of Health | R35 GM136261 | Aaron A Hoskins |
| National Science Foundation | Graduate Research Fellowship Program DGE-1747503 | Katherine A Senn |
| Helen Hay Whitney Foundation | | Max E Wilkinson |
| National Institutes of Health | T32 GM135066 | Katherine A Senn |

The funders had no role in study design, data collection and interpretation, or the decision to submit the work for publication.

### Author contributions

Katherine A Senn, Conceptualization, Resources, Data curation, Formal analysis, Investigation, Visualization, Writing – original draft, Writing – review and editing; Karli A Lipinski, Conceptualization, Resources, Data curation, Software, Formal analysis, Investigation, Visualization, Writing – original draft, Writing – review and editing; Natalie J Zeps, Formal analysis, Investigation; Amory F Griffin, Software, Formal analysis, Investigation; Max E Wilkinson, Conceptualization, Data curation, Formal analysis, Validation, Investigation, Visualization, Methodology, Writing – original draft, Project administration, Writing – review and editing; Aaron A Hoskins, Conceptualization, Formal analysis, Supervision, Funding acquisition, Investigation, Writing – original draft, Project administration, Writing – review and editing

### Author ORCIDs

Katherine A Senn ⬮ https://orcid.org/0000-0003-2172-7113
Aaron A Hoskins ⬮ https://orcid.org/0000-0002-9777-519X

Reviewer #1 (Public review): https://doi.org/10.7554/eLife.100449.3.sa1
Reviewer #2 (Public review): https://doi.org/10.7554/eLife.100449.3.sa2
Reviewer #3 (Public review): https://doi.org/10.7554/eLife.100449.3.sa3

Author response https://doi.org/10.7554/eLife.100449.3.sa4

## Additional files

### Supplementary files
- Supplementary file 1. Read mapping and other information for RNA-sequencing (RNA-seq) datasets.
- Supplementary file 2. RNA-sequencing (RNA-seq) datasets used for analysis for each figure.
- Supplementary file 3. ACT1-CUP1 reporter sequences with variable BS to 3' splice site (SS) distances.
- Supplementary file 4. Cryo-electron microscopy (cryo-EM) data collection and refinement statistics.
- Supplementary file 5. Mutations identified in *fyv6Δ* suppressor strains.
- Supplementary file 6. Yeast strains used in these studies.
- Supplementary file 7. Plasmids used in these studies.
- Supplementary file 8. Oligonucleotides used in these studies.
- MDAR checklist

### Data availability
RNA-seq data has been uploaded to NCBI SRA with BioProject ID PRJNA1113593. R code for calculation of FAnS values can be found at the following DOI: 10.5281/zenodo.13798743. The cryo-EM maps and atomic model of the P complex (state I) have been deposited in the Electron Microscopy Data Bank (https://www.ebi.ac.uk/pdbe/emdb/) and in the Protein Data Bank (https://www.rcsb.org). The data are available under the following accession codes: P complex state I (EMD-47157, EMD-47160, EMD-47161, EMD-47162, EMD47163, EMD-47164; PDB 9DTR), state II (EMD-47158), and state III (EMD-47159). Raw cryo-EM data is available from EMPIAR (EMPIAR-12330). Plasmids are available from Addgene. Yeast strains and other materials are freely available upon request. Other data generated or analyzed in the study are included in the manuscript, supporting files, or source data files.

The following datasets were generated:

| Author(s) | Year | Dataset title | Dataset URL | Database and Identifier |
|---|---|---|---|---|
| Senn KA, Lipinski KA, Zeps NJ, Griffin AF, Wilkinson MW, Hoskins AA | 2024 | Fyv6 Deletion RNA Sequencing | https://www.ncbi.nlm.nih.gov/bioproject/PRJNA1113593 | NCBI BioProject, PRJNA1113593 |
| Lipinski KA, Griffin AF, Hoskins AA | 2024 | Control of 3' splice site selection by the yeast splicing factor Fyv6 | https://doi.org/10.5281/zenodo.13798743 | Zenodo, 10.5281/zenodo.13798743 |
| Wilkinson ME, Hoskins AA | 2024 | Structure of the yeast post-catalytic P complex spliceosome at 2.3 Angstrom resolution | https://www.rcsb.org/structure/9DTR | RCSB Protein Data Bank, 9DTR |
| Wilkinson ME, Hoskins AA | 2024 | Yeast post-catalytic P complex spliceosome, state I, overall map | https://www.ebi.ac.uk/emdb/EMD-47157 | Electron Microscopy Data Bank, EMD-47157 |
| Wilkinson ME, Hoskins AA | 2024 | Yeast post-catalytic P complex spliceosome, focussed refinement on Prp22 | https://www.ebi.ac.uk/emdb/EMD-47160 | Electron Microscopy Data Bank, EMD-47160 |
| Wilkinson ME, Hoskins AA | 2024 | Yeast post-catalytic P complex spliceosome, focussed refinement on the U2 snRNP | https://www.ebi.ac.uk/emdb/EMD-47161 | Electron Microscopy Data Bank, EMD-47161 |

*Continued on next page*

*Continued*

| Author(s) | Year | Dataset title | Dataset URL | Database and Identifier |
|---|---|---|---|---|
| Wilkinson ME, Hoskins AA | 2024 | Yeast post-catalytic P complex spliceosome, focussed refinement on the U5 snRNP Sm ring | https://www.ebi.ac.uk/emdb/EMD-47162 | Electron Microscopy Data Bank, EMD-47162 |
| Wilkinson ME, Hoskins AA | 2024 | Yeast post-catalytic P complex spliceosome, focussed refinement on the NTC | https://www.ebi.ac.uk/emdb/EMD-47163 | Electron Microscopy Data Bank, EMD-47163 |
| Wilkinson ME, Hoskins AA | 2024 | Yeast post-catalytic P complex spliceosome, focussed refinement on the Cwc22 N-terminal domain | https://www.ebi.ac.uk/emdb/EMD-47164 | Electron Microscopy Data Bank, EMD-47164 |
| Wilkinson ME, Hoskins AA | 2024 | Yeast post-catalytic P complex spliceosome, state II, overall map | https://www.ebi.ac.uk/emdb/EMD-47158 | Electron Microscopy Data Bank, EMD-47158 |
| Wilkinson ME, Hoskins AA | 2024 | Yeast post-catalytic P complex spliceosome, state III, overall map | https://www.ebi.ac.uk/emdb/EMD-47159 | Electron Microscopy Data Bank, EMD-47159 |
| Wilkinson ME, Hoskins AA | 2024 | Yeast Post-catalytic P complex spliceosome | https://www.ebi.ac.uk/empiar/EMPIAR-12330 | Electron Microscopy Public Image Archive, EMPIAR-12330 |

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
