## [Editor Report · eLife Assessment]

This **important** study addresses how 3' splice site choice is modulated by the conserved spliceosome-associated protein Fyv6. The authors provide **compelling** evidence that Fyv6 functions to enable selection of 3' splice sites distal to a branch point and in doing so antagonizes more proximal, suboptimal 3' splice sites.

---

## [Referee Report · Reviewer #1 (Public review)]

Summary:

A key challenge at the second chemical step of splicing is the identification of the 3' splice site of an intron. This requires recruitment of factors dedicated to the second chemical step of splicing and exclusion of factors dedicated to the first chemical step of splicing. Through the highest resolution cyroEM structure of the spliceosome to-date, the authors show the binding site for Fyv6, a factor dedicated to the second chemical step of splicing, is mutually exclusive with the binding site for a distinct factor dedicated to the first chemical step of splicing, highlighting that splicing factors bind to the spliceosome at a specific stage not only by recognizing features specific to that stage but also by competing with factors that bind at other stages. The authors further reveal that Fyv6 functions at the second chemical step to promote selection of 3' splice sites distal to a branch point and thereby discriminate against proximal, suboptimal 3' splice site. Lastly, the authors show by cyroEM that Fyv6 physically interacts with the RNA helicase Prp22 and by genetics Fyv6 functionally interacts with this factor, implicating Fyv6 in 3'SS proofreading and mRNA release from the spliceosome. The evidence for this study is robust, with the inclusion of genomics, reporter assays, genetics, and cyroEM. Further, the data overall justify the conclusions, which will be of broad interest.

Strengths:

(1) The resolution of the cryoEM structure of Fyv6-bound spliceosomes at the second chemical step of splicing is exceptional (2.3 Angstroms at the catalytic core; 3.0-3.7 Angstroms at the periphery), providing the best view of this spliceosomal intermediate in particular and the core of the spliceosome in general.

(2) The authors observe by cryoEM three distinct states of this spliceosome, each distinguished from the next by progressive loss of protein factors and/or RNA residues. The authors appropriately refrain from overinterpreting these states as reflecting distinct states in the splicing cycle, as too many cyroEM studies are prone to do, and instead interpret these observations to suggest interdependencies of binding. For example, when Fyv6, Slu7, and Prp18 are not observed, neither are the first and second residues of the intron, which otherwise interact, suggesting an interdependence between 3' splice site docking on the 5' splice site and binding of these second step factors to the spliceosome.

(3) Conclusions are supported from multiple angles.

(4) The interaction between Fyv6 and Syf1, revealed by the cyroEM structure, was shown to account for the temperature-sensitive phenotypes of a fyv6 deletion, through a truncation analysis.

(5) Splicing changes were observed in vivo both by indirect copper reporter assays and directly by RT-PCR.

(6) Changes observed by RNA-seq are validated by RT-PCR.

(7) The authors go beyond simply observing a general shift to proximal 3'SS usage in the fyv6 deletion by RNA-seq by experimentally varying branch point to 3' splice site distance experimentally in a reporter and demonstrating in a controlled system that Fyv6 promotes distal 3' splice sites.

(8) The importance of the Fyv6-Syf1 interaction for 3'SS recognition is demonstrated by truncations of both Fyv6 and of Syf1.

(9) In general, the study was executed thoroughly and presented clearly.

Comments on revisions:

The authors have satisfactorily addressed the comments.

---

## [Referee Report · Reviewer #2 (Public review)]

In this manuscript, Senn, Lipinski, and colleagues report on the structure and function of the conserved spliceosomal protein Fyv6. Pre-mRNA splicing is a critical gene expression step that occurs in two steps, branching and exon ligation. Fyv6 had been recently identified by the Hoskins' lab as a factor that aids exon ligation (Lipinski et al., 2023), yet the mechanistic basis for Fyv6 function was less clear. Here, the authors combine yeast genetics, transcriptomics, biochemical assays, and structural biology to reveal the function of Fyv6. Specifically, they describe that Fyv6 promotes the usage of distal 3'SSs by stabilizing a network of interactions that include the RNA helicase PRP22 and the spliceosome subunit SYF1. They discuss a generalizible mechanism for splice site proofreading by spliceosomsal RNA helicases that could be modulated by other, regulatory splicing factors.

This is a very high quality study, which expertly combines various approaches to provide new insights into the regulation of 3'SS choice, docking, and undocking. The cryo-EM data is also of excellent quality, which substantially extends on previous yeast P complex structures. This is also supported by the authors use of the latest data analysis tools (Relion-5, AlphaFold2 multimer predictions, Modelangelo). The authors re-evaluate published EM densities of yeast spliceosome complexes (B*, C,C*,P) for the presence or absence of Fyv6, substantiate Fyv6 as a 2nd step specific factor, confirm it as the homolog of the human protein FAM192A, and provide a model for how Fyv6 may fit into the splicing pathway. The biochemical experiments on probing the splicing effects of BP to 3'SS distances after Fyv6 KO, genetic experiments to probe Fyv6 and Syf1 domains, and the suppressor screening add substantially to the study and are well executed. The manuscript is clearly written and we particularly appreciated the nuanced discussions, for example for an alternative model by which Prp22 influences 3'SS undocking. The research findings will be of great interest to the pre-mRNA splicing community.

Comments on revisions:

I'm satisfied with the changes.

---

## [Referee Report · Reviewer #3 (Public review)]

In this manuscript the authors expand their initial identification of Fyv6 as a protein involved in the second step of pre-mRNA splicing to investigate the transcriptome-wide impact of Fyv6 on splicing and gain a deeper understanding of the mechanism of Fyv6 action.

They first use deep sequencing of transcripts in cells depleted of Fyv6 together with Upf1 (to limit loss of mis-spliced transcripts) to identify broad changes in the transcriptome due to loss of Fyv6. This includes both changes in overall gene expression, that are not deeply discussed, as well as alterations in choice of 3' splice sites - which is the focus of the rest of the manuscript

They next provide the highest resolution structure of the post-catalytic spliceosome to date; providing unparalleled insight into details of the active site and peripheral components that haven't been well characterized previously.

Using this structure they identify functionally critical interactions of Fyv6 with Syf1 but not Prp22, Prp8 and Slu7. Finally, a suppressor screen additionally provides extensive new information regarding functional interactions between these second step factors.

Overall this manuscript reports new and essential information regarding molecular interactions within the spliceosome that determine the use of the 3' splice site. It would be helpful, especially to the non-expert, to summarize these in a table, figure or schematic in the discussion.

Comments on revisions:

I'm satisfied with the changes made in the revision.

---

## [Author Response]

The following is the authors’ response to the original reviews.

**eLife Assessment**
This important study addresses how 3' splice site choice is modulated by the conserved spliceosome-associated protein Fyv6. The authors provide compelling evidence Fyv6 functions to enable selection of 3' splice sites distal to a branch point and in doing so antagonizes more proximal, suboptimal 3' splice sites. The study would be improved through a more nuanced discussion of alternative possibilities and models, for instance in discussing the phenotypic impact of Fyv6 deletion.

We thank the editors and reviewers for their supportive comments and assessment of this manuscript. We have improved the discussion at several points as suggested by the reviewers to include discussion of alternative possibilities.

**Public Reviews:**

**Reviewer #1 (Public Review):**
Summary:A key challenge at the second chemical step of splicing is the identification of the 3' splice site of an intron. This requires recruitment of factors dedicated to the second chemical step of splicing and exclusion of factors dedicated to the first chemical step of splicing. Through the highest resolution cyroEM structure of the spliceosome to-date, the authors show the binding site for Fyv6, a factor dedicated to the second chemical step of splicing, is mutually exclusive with the binding site for a distinct factor dedicated to the first chemical step of splicing, highlighting that splicing factors bind to the spliceosome at a specific stage not only by recognizing features specific to that stage but also by competing with factors that bind at other stages. The authors further reveal that Fyv6 functions at the second chemical step to promote selection of 3' splice sites distal to a branch point and thereby discriminate against proximal, suboptimal 3' splice site. Lastly, the authors show by cyroEM that Fyv6 physically interacts with the RNA helicase Prp22 and by genetics Fyv6 functionally interacts with this factor, implicating Fyv6 in 3'SS proofreading and mRNA release from the spliceosome. The evidence for this study is robust, with the inclusion of genomics, reporter assays, genetics, and cyroEM. Further, the data overall justify the conclusions, which will be of broad interest.Strengths:(1) The resolution of the cryoEM structure of Fyv6-bound spliceosomes at the second chemical step of splicing is exceptional (2.3 Angstroms at the catalytic core; 3.0-3.7 Angstroms at the periphery), providing the best view of this spliceosomal intermediate in particular and the core of the spliceosome in general.(2) The authors observe by cryoEM three distinct states of this spliceosome, each distinguished from the next by progressive loss of protein factors and/or RNA residues. The authors appropriately refrain from overinterpreting these states as reflecting distinct states in the splicing cycle, as too many cyroEM studies are prone to do, and instead interpret these observations to suggest interdependencies of binding. For example, when Fyv6, Slu7, and Prp18 are not observed, neither are the first and second residues of the intron, which otherwise interact, suggesting an interdependence between 3' splice site docking on the 5' splice site and binding of these second step factors to the spliceosome.(3) Conclusions are supported from multiple angles.(4) The interaction between Fyv6 and Syf1, revealed by the cyroEM structure, was shown to account for the temperature-sensitive phenotypes of a fyv6 deletion, through a truncation analysis.(5) Splicing changes were observed in vivo both by indirect copper reporter assays and directly by RT-PCR.(6) Changes observed by RNA-seq are validated by RT-PCR.(7) The authors go beyond simply observing a general shift to proximal 3'SS usage in the fyv6 deletion by RNA-seq by experimentally varying branch point to 3' splice site distance experimentally in a reporter and demonstrating in a controlled system that Fyv6 promotes distal 3' splice sites.(8) The importance of the Fyv6-Syf1 interaction for 3'SS recognition is demonstrated by truncations of both Fyv6 and of Syf1.(9) In general, the study was executed thoroughly and presented clearly.

We thank the reviewer for their recognition of the strengths of our multi-faceted approach that led to highly supported conclusions.

Weaknesses:(1) Despite the authors restraint in interpreting the three states of the spliceosome observed by cyroEM as sequential intermediates along the splicing pathway, it would be helpful to the general reader to explicitly acknowledge the alternative possibility that the difference states simply reflect decomposition from one intermediate during isolation of the complex (i.e., the loss of protein is an in vitro artifact, if an informative one).

We thank the reviewer for noticing our restraint in interpreting these structures, and we agree that the scenario described by the reviewer is a possibility. We have now explicitly mentioned this in the Discussion on lines 755-757.

(2) The authors acknowledge that for prp8 suppressors of the fyv6 deletion, suppression may be indirect, as originally proposed by the Query and Konarska labs - that is, that defects in the second step conformation of the spliceosome can be indirectly suppressed by compensating, destabilizing mutations in the first step spliceosome. Whereas some of the other suppressors of the fyv6 deletion can be interpreted as impacting directly the second step spliceosome (e.g., because the gene product is only present in the second step conformation), it seems that many more suppressors beyond prp8 mutants, especially those corresponding to bulky substitutions, which would more likely destabilize than stabilize, could similarly act indirectly by destabilization of first step conformation. The authors should acknowledge this where appropriate (e.g., for factors like Prp8 that are present in both first and second step conformations).

We agree that this is also a possibility and have now included this on lines 480-486.

**Reviewer #2 (Public Review):**
In this manuscript, Senn, Lipinski, and colleagues report on the structure and function of the conserved spliceosomal protein Fyv6. Pre-mRNA splicing is a critical gene expression step that occurs in two steps, branching and exon ligation. Fyv6 had been recently identified by the Hoskins' lab as a factor that aids exon ligation (Lipinski et al., 2023), yet the mechanistic basis for Fyv6 function was less clear. Here, the authors combine yeast genetics, transcriptomics, biochemical assays, and structural biology to reveal the function of Fyv6. Specifically, they describe that Fyv6 promotes the usage of distal 3'SSs by stabilizing a network of interactions that include the RNA helicase PRP22 and the spliceosome subunit SYF1. They discuss a generalizible mechanism for splice site proofreading by spliceosomsal RNA helicases that could be modulated by other, regulatory splicing factors.This is a very high quality study, which expertly combines various approaches to provide new insights into the regulation of 3'SS choice, docking, and undocking. The cryo-EM data is also of excellent quality, which substantially extends on previous yeast P complex structures. This is also supported by the authors use of the latest data analysis tools (Relion-5, AlphaFold2 multimer predictions, Modelangelo). The authors re-evaluate published EM densities of yeast spliceosome complexes (B*, C,C*,P) for the presence or absence of Fyv6, substantiate Fyv6 as a 2nd step specific factor, confirm it as the homolog of the human protein FAM192A, and provide a model for how Fyv6 may fit into the splicing pathway. The biochemical experiments on probing the splicing effects of BP to 3'SS distances after Fyv6 KO, genetic experiments to probe Fyv6 and Syf1 domains, and the suppressor screening add substantially to the study and are well executed. The manuscript is clearly written and we particularly appreciated the nuanced discussions, for example for an alternative model by which Prp22 influences 3'SS undocking. The research findings will be of great interest to the pre-mRNA splicing community.

We thank the reviewer for their positive comments on our manuscript.

We have only few comments to improve an already strong manuscript.Comments:(1) Can the authors comment on how they justify K+ ion positions in their models (e.g. the K+ ion bridging G-1 and G+1 nucleotides)? How do they discriminate e.g. in the 'G-1 and G+1' case K+ from water?

The assignment of K+ at this position is justified by both longer coordination distances and relatively high cryo-EM density compared to structured water molecules in the same vicinity. We have added a panel to figure3-figure supplement 4C to show the density for the G-1/G+1 bridging K+ ion and to show the adjacent density for putative water molecules which coordinate the ion. The K+ ion density is larger and has stronger signal than the adjacent water molecules. The coordination distances are also longer than would be expected for a Mg2+. For these reasons and because K+ was present in the purification buffer, we modelled the density as K+.

(2) The authors comment on Yju2 and Fyv6 assignments in all yeast structures except for the ILS. Can the authors comment on if they have also looked into the assignment of Yju2 in the yeast ILS structure in the same manner? While it is possible that Fyv6 could dissociate and Yju2 reassociate at the P to ILS transition, this would merit a closer look given that in the yeast P complex Yju2 had been misassigned previously.

We thank the reviewer for pointing out this very interesting topic! We have used ModelAngelo to analyze the *S. cerevisiae* ILS structure for support of density assignment as Yju2 (and not Fyv6). This analysis supports the assignment as Yju2 in this structure and we have no evidence to doubt its presence in those particular purified spliceosomes. We have updated Figure 4- figure supplement 1B accordingly.

That being said, we do think that this issue should be studied more carefully in the future. The *S. cerevisiae* ILS structure (5Y88) was determined by purifying spliceosome complexes with a TAP-tag on Yju2. So the conclusion that Yju2 is part of the ILS spliceosome involves some circular logic: Yju2 is part of ILS spliceosome complexes because it is present in ILS complexes purified with Yju2. We also note that Yju2 was absent in ILS complexes recently determined from metazoans by the Plaschka group. We have added some additional nuance to the Discussion to raise this important mechanistic point at lines 711-718.

(3) For accessibility to a general reader, figures 1c, d, e, 2a, b, would benefit from additional headings or labels, to immediately convey what is being displayed. It is also not clear to us if Fig 1e might fit better in the supplement and be instead replaced by Supplementary Figure 1a (wt) , b (delta upf1), and a new c (delta fyv6) and new d (delta upf1, delta fyv6). This may allow the reader to better follow the rationale of the authors' use of the Fyv6/Upf1 double deletion.

We thank the reviewer for the suggestion and have updated Figures 1 C-E to include additional information in the headings and labels. We have not changed the labels in Figures 2A, B but have added additional clarifying language to the legend.

In terms of rearranging the figures, we thank the reviewer for the suggestion but have decided that the figures are best left in their current ordering.

(4) The authors carefully interpret the various suppressor mutants, yet to a general reader the authors may wish to focus this section on only the most critical mutants for a better flow of the text.

We thank the reviewer for this suggestion. While this section of the manuscript does contain (to quote Reviewer #3) “extensive new information regarding functional interactions”, it was a bit long. We have reduced this section of the manuscript by ~200 words for a more focused presentation for general readers.

**Reviewer #3 (Public Review):**
In this manuscript the authors expand their initial identification of Fyv6 as a protein involved in the second step of pre-mRNA splicing to investigate the transcriptome-wide impact of Fyv6 on splicing and gain a deeper understanding of the mechanism of Fyv6 action.They first use deep sequencing of transcripts in cells depleted of Fyv6 together with Upf1 (to limit loss of mis-spliced transcripts) to identify broad changes in the transcriptome due to loss of Fyv6. This includes both changes in overall gene expression, that are not deeply discussed, as well as alterations in choice of 3' splice sites - which is the focus of the rest of the manuscriptThey next provide the highest resolution structure of the post-catalytic spliceosome to date; providing unparalleled insight into details of the active site and peripheral components that haven't been well characterized previously.Using this structure they identify functionally critical interactions of Fyv6 with Syf1 but not Prp22, Prp8 and Slu7. Finally, a suppressor screen additionally provides extensive new information regarding functional interactions between these second step factors.Overall this manuscript reports new and essential information regarding molecular interactions within the spliceosome that determine the use of the 3' splice site. It would be helpful, especially to the non-expert, to summarize these in a table, figure or schematic in the discussion.

We thank the reviewer for the positive comments and suggestions. We did include a summary figure in panel 7H. However, it was a bit buried. To highlight the summary figure more clearly, we have moved panel 7H to its own figure (Fig. 8).

**Recommendations for the authors:**

**Reviewer #1 (Recommendations For The Authors):**
(1) The resolution of some panels is poor, nearly illegible (e.g., Supp Fig 1A, B).

The resolution of panels in supplemental figure 1 has been increased. However, this may be an artifact of the PDF conversion process. We will pay attention to this during the publication process.

(2) Panel S6B: 6HYU is a structure of DHX8, not DDX8

We have corrected DDX8 to DHX8 in Supplemental Fig. S6D and associated figure legend.

(3) The result that Syf1 truncations can suppress the Fyv6 deletion is impressive. The subsequent discussion seems muddled. A discussion of Fyv6 binding at the first step, instead of Yju2, doesn't seem relevant here (though worthy of consideration in the discussion), given that the starting mutation is the Fyv6 deletion. Further, conjuring rebinding of Yju2 based on the data in the paper seems unnecessarily speculative (assumes that biochemical state III is on pathway), unless I am unaware of some other evidence for such rebinding. Instead, a simpler explanation would seem to be that in the absence of Fyv6, Syf1 inappropriately binds Yju2 instead at the second step and that deletion of the common Fyv6/Yju2 binding site on Syf1 suppresses this defect. In this case, the ts phenotype of the Fyv6 deletion would result from inappropriate binding of Yju2, and the splicing defect would be due to loss of Fyv6 activity. Alternatively, especially considering the work of the labs of Query and Konarska, the authors should consider the possibility that (i) the Fyv6 deletion destabilizes the second step conformation, shifting an equilibrium to the first step conformation, and that (ii) the Syf1 truncation destabilizes binding of Yju2, thereby restoring the equilibrium. In this case the ts phenotype of the Fyv6 deletion is due to a disturbed equilibrium and the splicing defect is due to the failure of Fyv6 to function at the second step.

We believe the reviewer is specifically referencing the final paragraph of this Results section (the paragraph that comes just before the section “Mutations in many different splicing factors…”). In retrospect, we agree that our discussion was convoluted. In particular, we emphasized rebinding of Yju2 based on its presence in the cryo-EM structure of the yeast ILS complex. However, given some uncertainties about whether or not Yju2 is a *bona fide* ILS component (as discussed above). We don’t think it is appropriate to over-emphasize rebinding of Yju2 and have decided to incorporate the elegant mechanisms proposed by the reviewer. This paragraph has now been edited accordingly (lines 386-395).

(4) The authors imply they have performed biochemical studies, which I think is misleading. Of course, RT-PCR and primer extension assays for example are performed in vitro, but these are an analysis of RNA events that occurred in vivo. In my view a higher threshold should be used for defining "biochemistry". To me "biochemistry" would imply that the authors have, for example, investigated 3' splice site usage in splicing extracts of the fyv6 deletion or engaged in an analysis of the Syf1-Fyv6 interaction involving the expression of the interacting domains in bacteria followed by a binding analysis in the test tube.

We disagree with the reviewer on this point. Biochemistry is defined as the “branch of sciences concerned with the chemical substances, reactions, and physico chemical processes which occur within living organisms; biological or physical chemistry.” (Oxford English Dictionary). Biochemical studies are not defined by whether or not they take place in vitro, in vivo, or even in silico. Indeed, much of the history of biochemistry (especially in studies of metabolism, for example) involved experiments occurring in vivo that reported on the molecular properties and mechanisms of biological processes. We think many of our experiments fall into this category including our structure/function analysis of splicing factors and the use of the ACT1-CUP1 reporter substrate.

(5) The monovalents are shown; inositol phosphate is shown; is the binding of Prp22 to RNA shown?

We have added a panel to Figure 3-figure supplement 4D showing density for the 3' exon within Prp22.

(6) The authors invoke undocking of the 3'SS in the P complex. Where is the 3'SS in the ILS? The author's model predicts: undocked.

In all ILS structures to date, the 3′ SS is undocked, in agreement with this prediction. We have now noted this observation in line 760.

(7) Would be helpful to show fyv6 deletion in Fig 1b.

We have included growth data for an additional fyv6 deletion strain (in a *cup1Δ* background) in Figure 1b. The results are quite similar to the upf1_Δ_ background except with slightly worse growth at 23°C.

**Reviewer #2 (Recommendations For The Authors):**
Minor comments(1) Fig.3b is the arrow indicating the right rotation?

This typo has been fixed.

(2) Fig.4b, panel H is annotated, which should read 'F'.

This typo has been fixed.

(3) Line 178: "Finally, we analyzed the sequence features of the alternative 3ʹ SS activated by loss of Fyv6." We would suggest 'used after' instead of 'activated by'.

We have replaced ‘activated by’ with ‘with increased use after’.

(4) In Line 544, the authors speculate on a Slu7 requirement for 3'SS docking and on 3'SS docking maintenance. In the results section (Line 265) they however only mention the latter possibility. These statements should be consistent.

We thank the reviewer for pointing this out. We have added a reference to docking maintenance to the results section at line 325.

(5) Line 476: "Unexpectedly, Prp22 I1133R was actually deleterious when Fyv6 was present for this reporter." We suggest removing "actually".

We have removed ‘actually’.

(6) The authors describe the observed changes in splicing events in absolute numbers (e.g. in Fig 1c). To better assess for the reader whether these numbers reflect large or small effects of Fyv6 in defining mRNA isoforms, it would be more useful to state these as percent changes of total events or to provide a reference number for how many introns are spliced in S.c. See for example the statements in Lines 132 and 145.

We have added a percentage at line 138 that indicates ~20% of introns in yeast showed splicing changes.

**Reviewer #3 (Recommendations For The Authors):**
Do the authors have a proposed explanation for the observed DGE in non-intron containing genes in the Fyv6 depleted cells?

The simplest explanation is that this is an indirect effect due to splicing changes occurring in other genes (such as transcription factors, ribosomal protein genes, etc..). It is possible that this can be further dissected in the future using shorter-term knockdown of Fyv6 using Anchors Away or AID-tagging. However, that is beyond the scope of the current manuscript, and we do not wish to comment on these non-intron containing genes further at present.

Figure 2A - What is going on with the events that show no FAnS value under one condition (i.e. are up against the X or Y axis)? These are of interest as most on the Y- axis are blue.

The events along one of the axes denote alternative splice sites that are only detected under one condition (either when Fyv6 is present or when it is absent). At this stage, we do not wish to interpret these events further since most have a relatively low number of reads overall.